# Hantaan virus-derived peptides that stabilize HLA-E could abrogate inhibition of CD56dimNKG2A+ NK cells

Manling Xue[1], Kang Tang[1], Yusi Zhang[1], Xiaoyue Xu[1], Chunmei Zhang[1], Jiajia Zuo[1], Fenglan Wang[2], Xiyue Zhang[3], Xuyang Zheng[4], Ran Zhuang[1], Yun Zhang[1], Boquan Jin[1], Ying Ma[1]*

1 Department of Immunology, Fourth Military Medical University, Xi'an, China, 2 Department of Infectious Disease, Eighth Hospital of Xi'an, Xi'an, China, 3 Department of Transfusion Medicine, Tangdu Hospital, Fourth Military Medical University, Xi'an, China, 4 Center for Infectious Diseases, Tangdu Hospital, Fourth Military Medical University, Xi'an, China

* mayingying@fmmu.edu.cn

## Abstract

NK cells could participate in the pathogenesis process of virus infectious diseases through the inhibitory receptor CD94/NKG2A interacting with HLA-E/virus-derived peptide complex. However, the effects and mechanisms of NKG2A-HLA-E axis-mediated NK cell responses in hemorrhagic fever with renal syndrome (HFRS) caused by Hantaan virus (HTNV) infection remain unclear. Single-cell RNA sequencing and flow cytometry were employed to analyze the phenotype and function of different NK cell subsets in HFRS patients. The K562/HLA-E cells binding assay was used for peptide affinity detection. The binding capacity of HLA-E/peptide-CD94/NKG2A was detected using ligand-receptor binding assay and tetramer staining. The cytotoxicity assay of NK cells against peptide-pulsed K562/HLA-E cells was conducted for functional evaluation. In this study, CD56dimCD16+NKG2A+ NK cells were the main subset in HFRS patients, showing activation and proliferation phenotypes with NKG2C-CD57- and the ability to secrete tumor necrosis factor-α (TNF-α), interferon-γ (IFN-γ) and cytotoxic mediators. Notably, none of the four identified HTNV epitopes presented by HLA-E could be recognized by CD94/NKG2A on CD56dimNKG2A+ NK cells. Furthermore, the subset of CD56dimNKG2A+ NK cells showed the enhanced cytolytic capacity against HTNV peptide pulsed K562/HLA-E cells *ex vivo*. Taken together, the findings demonstrate that HTNV-derived peptides presented by HLA-E could "abrogate" the inhibition of CD56dimNKG2A+ NK cells, contributing to the antiviral immune response in HFRS patients.

**Data availability statement:** The authors declare that all data supporting the findings of this study are available within this article and its Supplementary Information files. Single-cell RNAseq gene expression data have been deposited in the Gene Expression Omnibus database (GSE161354).

**Funding:** This work was supported by the National Natural Science Foundation of China (82341071 to YM, 81871239 to YM, 82272331 to YZ [Yun Zhang]) and the Key Research and Development Program of Shaanxi Province (2023-JC-ZD-49 to YM, 2022JZ-45 to CZ, 2023-JC-YB-640 to YZ [Yusi Zhang] and 2024SF-YBXM-286 to KT). The funders had no role in study design, data collection and analysis, decision to publish, or preparation of the manuscript.

**Competing interests:** The authors have declared that no competing interests exist.

## Author summary

Hantaan virus (HTNV) is one of the main pathogens causing hemorrhagic fever with renal syndrome (HFRS) characterized by fever, hemorrhage, renal injury, and thrombocytopenia. Recently, the studies have shown that the interaction of human leukocyte antigen E (HLA-E) and natural-killer group 2, member A (NK-G2A) inhibitory receptors could regulate the functions of NK cells, participating the pathogenesis process of virus infectious diseases. However, the role of NK cell response induced by HTNV infection in the pathogenesis of HFRS has not been completely determined. Here, the findings suggest a potential link between NKG2A-expressing NK cell functional profiles and HTNV infection. The link may be mediated by the mechanism of the lack of recognition between CD94/NKG2A and the HLA-E/HTNV peptide complex. This study may provide the mechanisms of NKG2A-HLA-E axis on regulating NK cell responses in HTNV infections.

## Introduction

Hantaviruses, significant class of zoonotic pathogens, are enveloped RNA viruses with negative-stranded genomes that belong to the *Bunyaviridae* family [1]. The viruses are mainly transmitted to humans in aerosols of virus-containing rodent excreta, such as saliva, urine, and faece [2]. Hantaviruses infection in humans can result in two kinds of acute and severe infectious diseases: hemorrhagic fever with renal syndrome (HFRS) in Eurasia and hantavirus cardiopulmonary syndrome (HCPS) in the Americas [3]. HFRS could be caused by the members of hantaviruses including Hantaan virus (HTNV), Puumala virus (PUUV), Dobrava virus (DOBV), and Seoul virus, while HCPS is mainly caused by Sin Nombre virus (SNV) and Andes virus (ANDV) [4]. Annually, about 200,000 cases of HFRS were reported worldwide, with more than 90% of the total cases reported in China [5]. HTNV is one of the main pathogens causing HFRS in China characterized by fever, hemorrhage, hypotensive shock, acute renal injury, and thrombocytopenia [6]. According to the report of Chinese Center for Disease Control and Prevention, a total of 98,849 HFRS cases with 649 fatalities were reported between 2013 and 2022 in China. At present, there is no approved etiological specific treatment strategy for HFRS in clinical. Because of the high morbidity and mortality, and poorly understood immunopathogenesis of HFRS, it is necessary to explore the immune mechanism against HTNV infection and develop effective antiviral or immunomodulatory therapy to avoid the potential serious outbreaks of this zoonotic disease in the future.

The efforts of immune system to defend against HTNV are important for clearance of the infection. Natural killer (NK) cells, known as innate cytotoxic lymphocytes in the immune system, play a crucial role in the early defense against hantaviruses. The circulating NK cells could be divided into two major subgroups based on the relative surface expression of CD56 and CD16: the mature $CD56^{dim}CD16^{+}$ NK cell subset, which represents the predominant NK cell subset with cytotoxic effects, and the immature $CD56^{bright}CD16^{-}$ NK cell subset, which shows limited cytotoxicity but efficient cytokines producing [7–9]. Both the two

subsets of NK cells could interact with virus and play functional roles in viral infections. In PUUV infected patients, the frequency of CD56dimCD16+ NK cells subset increased markedly, whereas the frequency of CD56brightCD16- NK cells subset decreased after symptom onset [10]. Moreover, a rapid expansion of activated cytotoxic CD56dim NK cell subset in the peripheral blood of PUUV-infected patients with an increased expression of the cytolytic proteins perforin and granzyme B was also observed [11].

The function of NK cell is regulated by the signal balance between a series of activating and inhibitory receptors expressed on the surface of NK cells, among which, the C-type lectin family members NKG2A, NKG2C and NKG2E are important receptors for the function of NK cells. NKG2A/C/E always form a heterodimer with CD94 expressed on both NK cells and activated CD8+ T cells. NKG2A has two immunoreceptor tyrosine-based inhibition motifs (ITIMs) in cytoplasmic region, which can recruit intracellular phosphatase SHP-1 as well as SHP-2. Upon binding to the ligand human leukocyte antigen-E (HLA-E) molecule in human, NKG2A could induce an inhibitory signal for both CD8+ T cells and NK cells [12,13]. HLA-E is a non-classical HLA class I (HLA Ib) molecule, which ubiquitously expressed on the surface of almost all types of cells. HLA-E displays limited polymorphism with only two functional alleles, HLA-E*0101 and HLA-E*0103, covering over 99% of human populations [14]. Normally, HLA-E presents peptides derived from the leading sequence of classical HLA class I (HLA-Ia) molecules (HLA-A, B, C) and HLA-G molecules, with the typically conserved sequence VMAPRT(V/L)(L/V/I/F)L. Under pathologic conditions, HLA-E could be overexpressed and present pathogen-derived peptides to be recognized by TCR on CD8+ T cells to induce effective CD8+ T cell responses against infection, or interact with CD94/NKG2A on both activated CD8+ T cells and NK cells, producing inhibition to CD8+ T cells and NK cell and contributing to viral immune evasion [15].

HLA-E-NKG2A interaction has been found to be involved in the pathogenesis of many infectious diseases. The hepatitis C virus (HCV) core protein derived-peptide (aa35-aa44) could be presented by HLA-E and then interacted with NKG2A/CD94, resulting in NK cell-mediated immune escape [16]. In patients with chronic hepatitis B (CHB), the upregulated expression of NKG2A on NK cells was observed with the impairment of function in cytokine secretion [17]. Blocking the interaction between NKG2A and HLA-E could enhance the cytotoxicity of NK cells from CHB patients. In severe acute respiratory syndrome coronavirus 2 (SARS-CoV-2) infected patients, the enhanced expression of NKG2A was observed on NK cells. The cytotoxicity and degranulation of NK cells from SARS-CoV-2 patients could be enhanced when NKG2A was blocked [18]. Thus, reducing the inhibitory signal produced by HLA-E-NKG2A axis may augment antiviral functions of NK cells [19]. However, some pathogen-derived peptides could be formed a complex with HLA-E but unable to be recognized and bound with CD94/NKG2A. For example, a peptide derived from SARS-CoV-2 non-structural protein 13 could be presented by HLA-E, but failed to be recognized with CD94/NKG2A on NK cells, rendering target cells more susceptible to NK cells attack [20]. Therefore, the interaction between NKG2A/CD94 and HLA-E/peptide complex is closely correlated with the effects of NK cells in protection against viral infections or mediating immune escape. Whether the HLA-E/virus-derived peptide complex could be recognized by CD94/NKG2A might be the key factor to determine the function of NK cells in virus infection. However, it still remains unknown about the role and mechanism of NK cell immune response mediated by HLA-E-NKG2A axis during HTNV infection.

In this study, we analyzed the functional subsets of NK cells during HTNV infection in HFRS patients, and investigated the role of HLA-E-NKG2A axis in NK cell response by detecting the phenotype and function of peripheral CD56dimNKG2A+ NK cells in HTNV infected patients. The peptides on HTNV that could be formed a complex with HLA-E were also identified. The characteristics of potential interactions between CD94/NKG2A and HLA-E/HTNV-derived peptide was also explored. The results may partially point out the mechanisms of NKG2A-HLA-E axis on regulating the function of NK cell in HFRS patients, which may provide a foundation for further study of the immunopathogenesis of HFRS after HTNV infection.

## Results

### Six clusters of peripheral blood NK cells were identified in HFRS patients

To profile the NK cell immune response to HTNV in peripheral blood of HFRS patients, we performed single-cell RNA sequencing (scRNA-Seq) analysis of peripheral blood mononuclear cells (PBMCs) from six HFRS patients and two uninfected controls. The clinical information of the samples used for sequencing were described in Table 1. According to

**Table 1. Information of subjects enrolled in scRNA-seq.**

| Sample No. | Age | Sex | Disease phase | Disease severity | NK cell counts/total sequencing cell counts |
|---|---|---|---|---|---|
| P04 | 52 | Male | Oliguricia | critical | 115/4556 |
| P06 | 34 | Male | Oliguricia | moderate | 62/3746 |
| P07 | 49 | Male | Fever | moderate | 73/4218 |
| P15 | 29 | Male | Fever, shock, and oliguricia | critical | 91/4551 |
| P16 | 37 | Male | Fever | moderate | 139/4129 |
| P26 | 26 | Male | Fever and shock | critical | 46/3763 |
| NC1 | 29 | Male | / | / | 718/4253 |
| NC2 | 33 | Female | / | / | 145/3660 |

P: patient of HFRS; NC: Uninfected control.

t-distributed stochastic neighbor embedding (t-SNE) algorithm, PBMCs were divided into 19 clusters (Fig 1A). The clusters 0, 1, 3, 4, 5, 8, 11, 12, 17 of PBMCs were designated as T/NK/NKT cells based on canonical markers reported previously [21]. The T/NK/NKT cells were then subdivided into 15 cell clusters (Clusters 0-14), among which, the cluster 5 expressed *CD3D*, *FCER1G*, *NCAM1*, *NCR1*, *PTPRC* and *TYROBP* was identified as NK cells in t-SNE plot (Fig 1B). Then NK cells were filtered and standardized into six cell clusters (Clusters 0-5) by t-SNE analysis.

Next, we analyzed the distribution of the three clusters of NK cells across different HFRS disease severities and compared with uninfected controls. The distribution pattern in the t-SNE plots and the bar graphs showed that NK cells from HFRS patients were mainly from cluster 0, 4 and 5 (Fig 1C and 1D). Specifically, cluster 0 and 5 of NK cells were enriched in HFRS patients with critical disease severity, the proportion of which were higher than that in moderate patients (Fig 1E and 1F). Taken together, six subclusters of peripheral blood NK cells were found in HFRS patients, among which, 0 and 5 clusters of NK cells may be associated with more severe HFRS.

### Two distinct subclusters of NK cells exhibited high expression of NKG2A and displayed signatures associated with inflammation and proliferation during HTNV infection

To further discriminate the three subclusters (cluster 0, 4 and 5) in the high-expression NK clusters of HFRS patients, the genes that represent the fundamental characteristic and phenotype of NK cells were analyzed (Fig 2A). When compared to the cluster 0, both the cluster 4 and 5 exhibited relatively high expression of genes *FCGR3A* (CD16) and *KLRC1* (NKG2A), a medium-level expression of gene *NCAM1* (CD56), a low level of gene *KLRC2* (NKG2C) and no expression of gene *B3GAT1* (CD57). Hence, the subclusters 4 and 5 of NK cells might mainly present a phenotype of CD56$^{dim}$CD16$^+$NKG2A$^+$NKG2C$^{lo}$CD57$^-$. Meanwhile, we compared the expression of above genes between HFRS patients and uninfected controls. Results showed that total NK cells of HFRS patients mainly expressed *NCAM1*, *B3GAT1*, *KLRC1* and *KLRC2* genes (S1A Fig). There was higher expression of *NCAM1*, *KLRC1* and *KLRC2* genes in moderate than those in critical HFRS patients and uninfected controls, but lower expression of *FCGR3A* gene than that in uninfected controls. While there was a significantly higher expression of *B3GAT1* gene than that in uninfected controls (S1B Fig).

Next, we investigated the function-related gene expression profile of cluster 4 and 5 in NK cells (Fig 2B). It was notable that cluster 4 NK cells in moderate HFRS patients mainly presented a high expression of genes such as *FOS*, *AREG*, *TNFAIP*, *DUSP2*, *NFKBIA* and *CD69*, indicating TNF signaling and the activation of NK cells. The cluster 5 NK cells in critical HFRS patients mainly showed a relatively high expression of genes such as *MKI67*, *STMN1*, *HMGB2*, *TUBA1B* and *TUBB*, which might be related to the proliferation ability and the cell cycle of NK cells. Pathway analysis also demonstrated that the cluster 4 NK cells was mainly activated in the TNF signaling pathway, while the cluster 5 NK cells mainly participated in the process of cell proliferation, differentiation and development (Fig 2C).

PLOS Pathogens

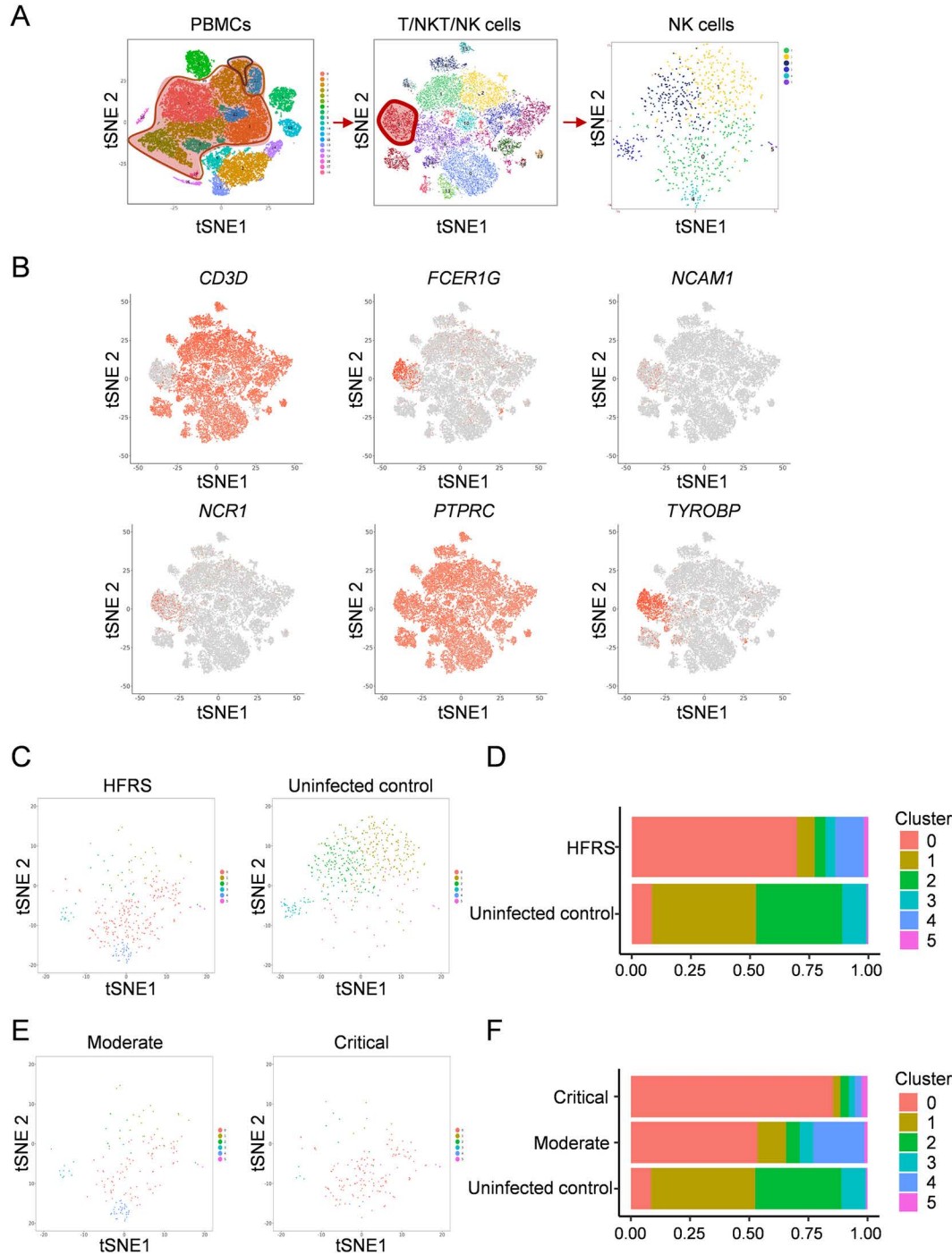

**Fig 1. ScRNA-seq analysis of peripheral blood NK cells in HFRS patients and uninfected control samples.** (A) The t-SNE presentation of the cell clusters within PBMC, T/NKT/NK cells and NK cells respectively from HFRS patients. (B)The t-SNE plots showed the signature genes (*CD3D*, *FCER1G*, *NCAM1*, *NCR1*, *PTPRC* and *TYROBP*) of NK cells in T/NKT/NK cells. (C) The t-SNE plots and (D) bar graphs of NK cells in HFRS patients with different disease severities and uninfected controls. (E) The t-SNE plots and (F) bar graphs of the proportion of NK clusters across HFRS patients with different disease severities and uninfected controls.

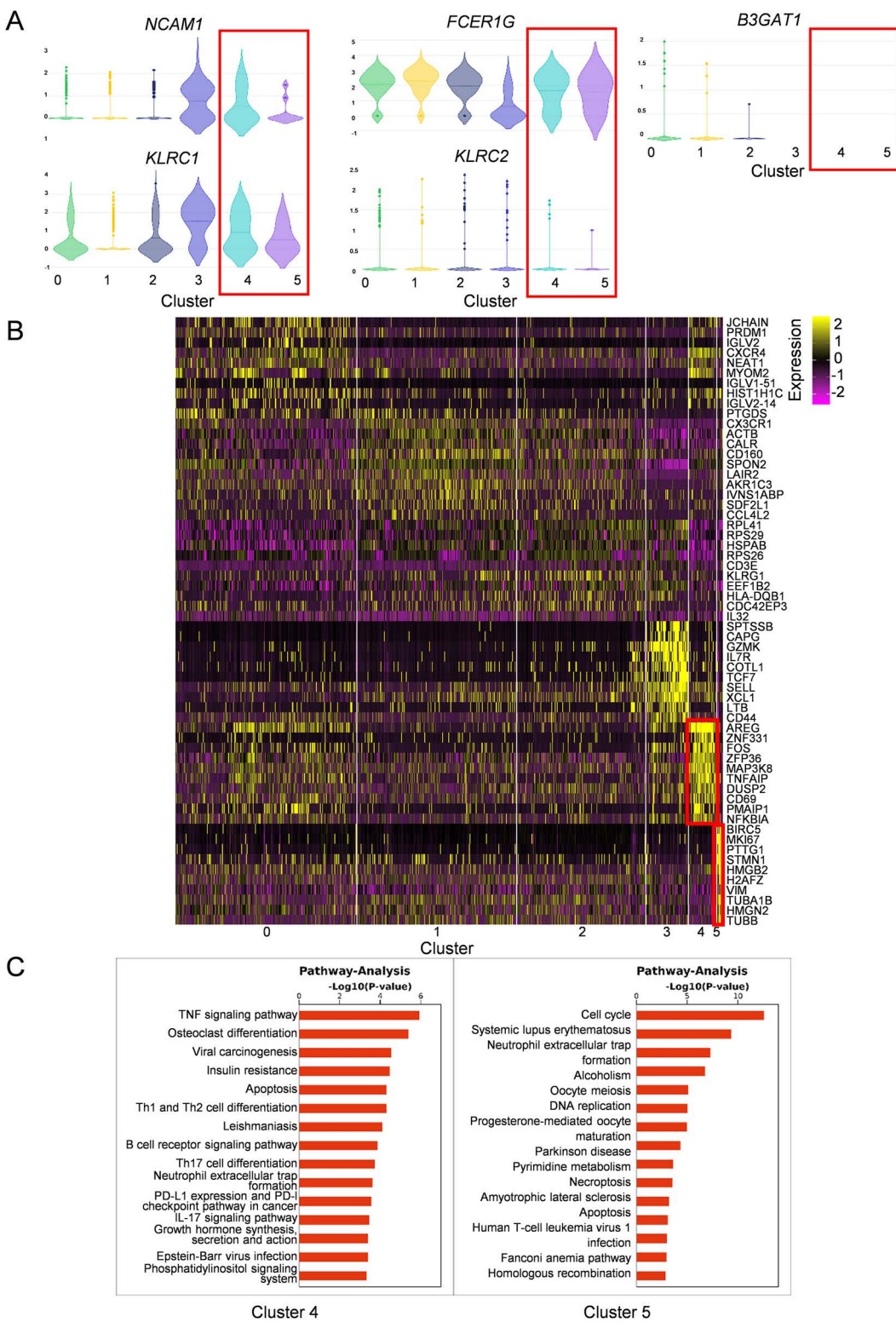

**Fig 2. The expression of representative genes in different subclusters of NK cells.** (A) The violin graph showing the comparison of representative genes (CD56/*NCAM1*, CD16/*FCER1G*, NKG2A/*KLRC1*, NKG2C/*KLRC2*, CD94/*KLRD1* and CD57/*B3GAT1*) expression in each cluster of NK cells. (B) The heatmap presenting gene expression profiles of each cluster in NK cells. (C) The pathway analysis showing the pathway activation in cluster 4 and 5 of NK cells.

**The frequency of NK cells with the phenotype CD56^dimCD16^+NKG2A^+NKG2C^-CD57^- increased in the peripheral blood of HFRS patients**

To confirm the change of the subclusters of NK cells in HTNV infection, we carried out flow cytometry (FCM) analysis on the frequencies and phenotypes of NK cells in PBMCs of HFRS patients. The results showed that the frequencies of CD56^dimCD16^+ NK cells in HFRS patients were lower than that of uninfected controls ($p<0.01$) (Fig 3A and 3B). The expression of NKG2A on CD56^dimCD16^+ NK cells showed that both the frequencies and mean fluorescence intensity (MFI) of NKG2A were significantly higher in HFRS patients than in uninfected controls ($p<0.05$ and $p<0.01$, respectively); whereas there was no difference in the expression of NKG2C on CD56^dimCD16^+ NK cells for both frequency and MFI between HFRS patients and uninfected controls (Fig 3C-E). Moreover, both the frequencies and the MFI of CD56^dimCD16^+NKG2A^+ NK cells were higher than that of CD56^dimCD16^+NKG2C^+ NK cells in HFRS patients ($p<0.01$ and $p<0.001$, respectively) (Fig 3F). Meanwhile, the expression of CD57 on CD56^dimCD16^+NKG2A^+NKG2C^- NK cells of HFRS patients were analyzed (Fig 3G), the result of which showed that the frequencies of CD57^- NK cells were higher than that of CD57^+ NK cells in HFRS patients ($p<0.001$) (Fig 3H). The gating strategy for CD56^dimCD16^+NKG2A^+NKG2C^-CD57^- NK cells (referred to as CD56^dimNKG2A^+ NK cells for short in following) was showed in S2 Fig.

**The CD56^dimNKG2A^+ NK cells were highly activated in HFRS patients**

Next, the activation status of CD56^dimNKG2A^+ NK cells in HFRS patients were detected. Notably, the frequencies of HLA-DR^+CD56^dimNKG2A^+ NK cells in both mild/moderate and severe/critical HFRS patients were significantly higher than that in uninfected controls ($p<0.01$ and $p<0.01$, respectively) (Fig 4A and 4B). The frequencies of CD69^+CD56^dimNKG2A^+ NK cells seem to be higher in both mild/moderate and severe/critical HFRS patients than that in uninfected controls, but showed no statistical difference among them. When comparing patients at different stages of the disease, the frequency of CD69^+CD56^dimNKG2A^+ NK cells was especially higher at convalescent stage than that at acute stage ($p<0.01$) (Fig 4C and 4D). The frequencies of both HLA-DR^+CD56^dimNKG2A^+ NK cells and CD69^+CD56^dimNKG2A^+ NK cells were higher at acute stage or convalescent stage of HFRS patients than that in uninfected controls ($p<0.01$ and $p<0.001$, respectively).

**CD56^dimNKG2A^+ NK cells of HFRS patients with severe/critical severity or at the acute stage exhibited a high proliferating capacity**

As focused on CD56^dimNKG2A^+ NK cells, the expression of proliferating marker Ki-67 was detected for the subset in peripheral blood of HFRS patients. The results showed that the frequency of Ki-67^+ cells in CD56^dimNKG2A^+ NK cells was higher in severe/critical patients than that in mild/moderate patients as well as than that in uninfected controls ($p<0.01$ and $p<0.001$, respectively) (Fig 5A and 5B). When compared between different stages of HFRS patients, the frequency of Ki-67^+ cells in CD56^dimNKG2A^+ NK cells was higher at acute stage than that at the convalescent stage and the uninfected controls ($p<0.001$ and $p<0.001$, respectively) (Fig 5C and 5D).

**CD56^dimNKG2A^+ NK Cells in HFRS Patients Correlate with Disease Severity and Clinical Course**

We investigated the production of TNF-α and IFN-γ by CD56^dimNKG2A^+NK cells in HFRS patients. The results showed that the frequencies of TNF-α^+CD56^dimNKG2A^+NK cells were significantly higher in severe/critical and mild/moderate HFRS patients compared to uninfected controls ($p<0.05$ and $p<0.05$, respectively) (Fig 6A and 6B). Similarly, the levels of IFN-γ produced by CD56^dimNKG2A^+ NK cells were significantly elevated in both severe/critical and mild/moderate HFRS patients compared to uninfected controls ($p<0.001$ and $p<0.05$, respectively) (Fig 6C and 6D). Furthermore, during the acute and convalescent stages, HFRS patients exhibited higher frequencies of TNF-α^+CD56^dimNKG2A^+ NK cells compared to uninfected controls ($p<0.01$ and $p<0.001$, respectively) (Fig 6G and 6H). Notably, the production of IFN-γ by CD56^dimNKG2A^+ NK cells was higher in HFRS patients during the acute stage compared to those in the convalescent stage and uninfected controls ($p<0.05$ and $p<0.001$, respectively) (Fig 6I and 6J). The frequency of TNF-α^+IFN-γ^+ CD56^dimNKG2A^+ NK cells

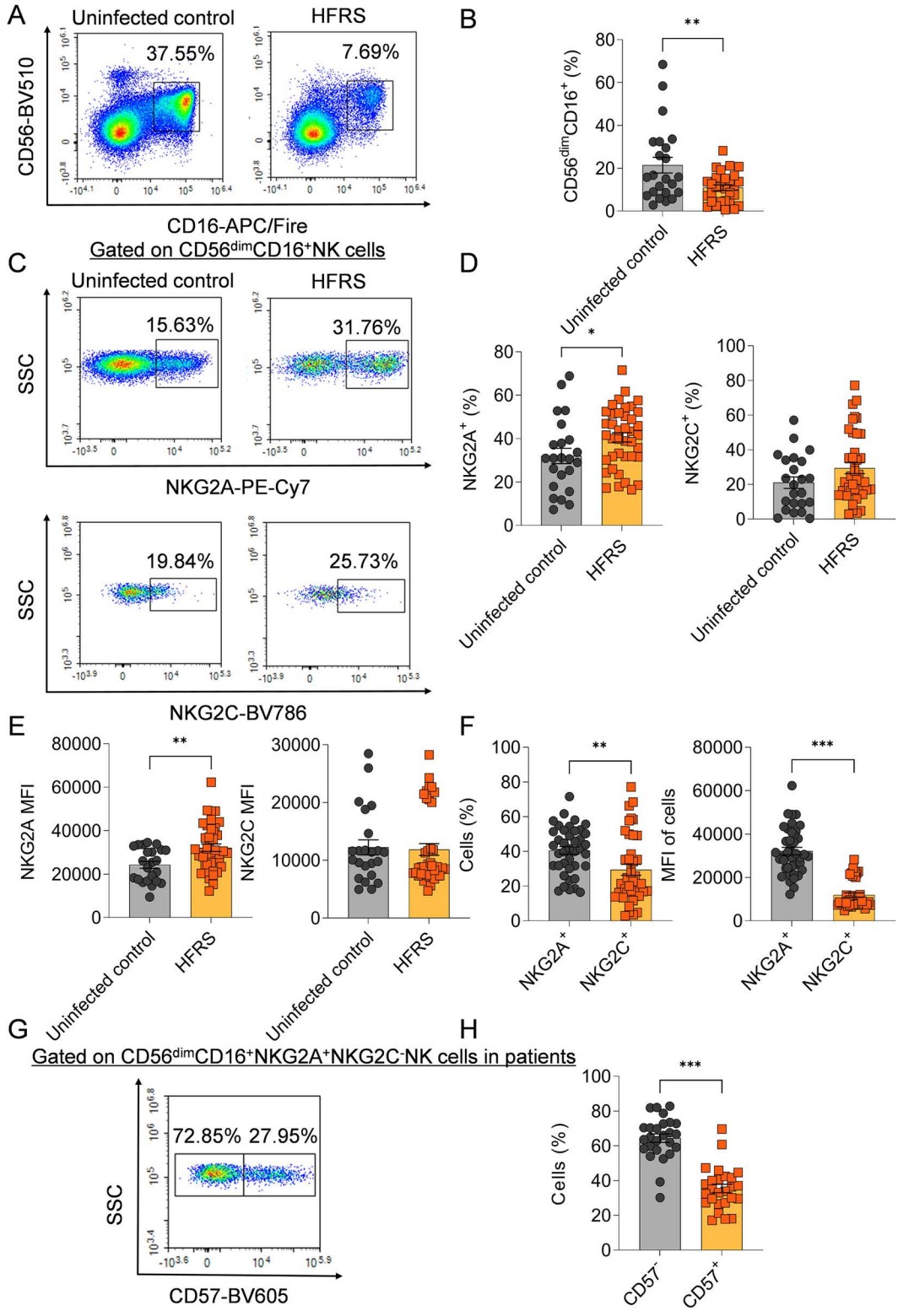

**Fig 3. The frequencies and phenotypic characteristics of CD56<sup>dim</sup> NK cells in HFRS patients.** (A) The representative flow cytometric plots and (B) the comparison of the frequencies of CD56<sup>dim</sup>CD16<sup>+</sup> NK cells between patients and uninfected controls. (C) The representative histogram figures and the comparison of (D) the frequencies as well as (E) the mean fluorescence intensity (MFI) of NKG2A or NKG2C gated on CD56<sup>dim</sup>CD16<sup>+</sup> NK cells between

patients and uninfected controls respectively. (F) The comparison of the frequencies or MFI between NKG2A+ and NKG2C+ cells gated on CD56dimCD16+ NK cells in HFRS patients. (G) The representative flow cytometric graph for the expression of CD57 in CD56dimCD16+NKG2A+NKG2C- NK cells. (H) The comparison of the frequencies between CD57- and CD57+ cells gated on CD56dimCD16+NKG2A+NKG2C- NK cells of HFRS patients. Statistical analysis was performed using the Mann-Whitney $U$ test. $p$-values less than 0.05 were considered statistically significant ($p < 0.05$, *; $p < 0.01$, **; $p < 0.001$, ***).

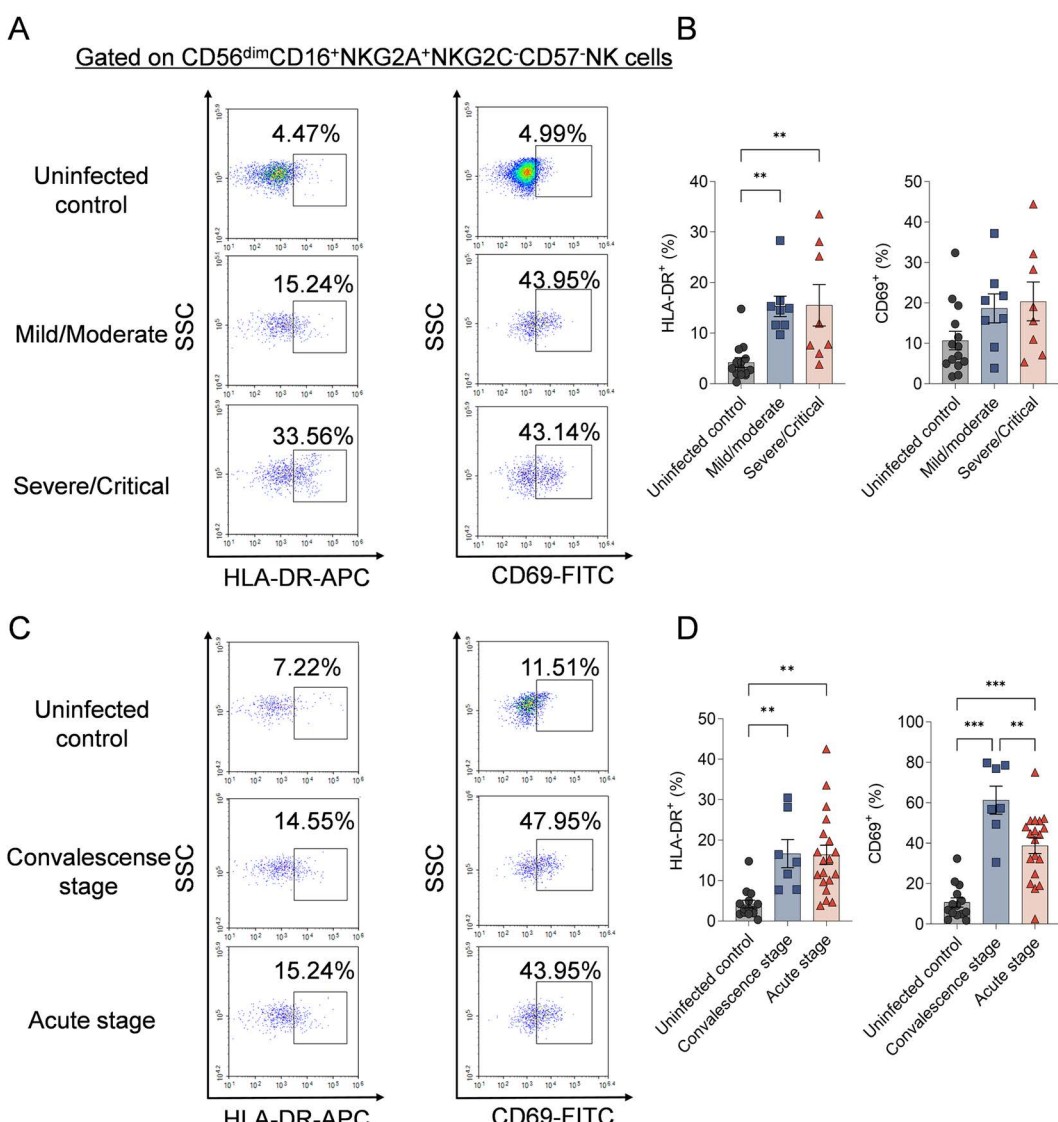

**Fig 4. The activation status of CD56dimCD16+NKG2A+NKG2C-CD57- NK cells in HFRS patients.** (A) Representative flow cytometric graphs and (B) the comparison of the frequencies of HLA-DR+ or CD69+ cells gated on CD56dimCD16+NKG2A+NKG2C-CD57- NK cells among HFRS patients with different disease severities and uninfected controls. (C) Representative flow cytometric plots and (D) the comparison of the frequencies of HLA-DR+ or CD69+ cells gated on CD56dimCD16+NKG2A+NKG2C-CD57- NK cells among HFRS patients at different stages and uninfected controls. Statistical analysis was performed using the Mann-Whitney $U$ test. $p$-values below 0.05 were considered statistically significant ($p < 0.01$, **; $p < 0.001$, ***).

was also higher during the acute stage compared to the convalescent stage and uninfected controls ($p < 0.05$ and $p < 0.01$, respectively) (Fig 6K and 6L), although no significant differences were observed between different disease severity groups (Fig 6E and 6F).

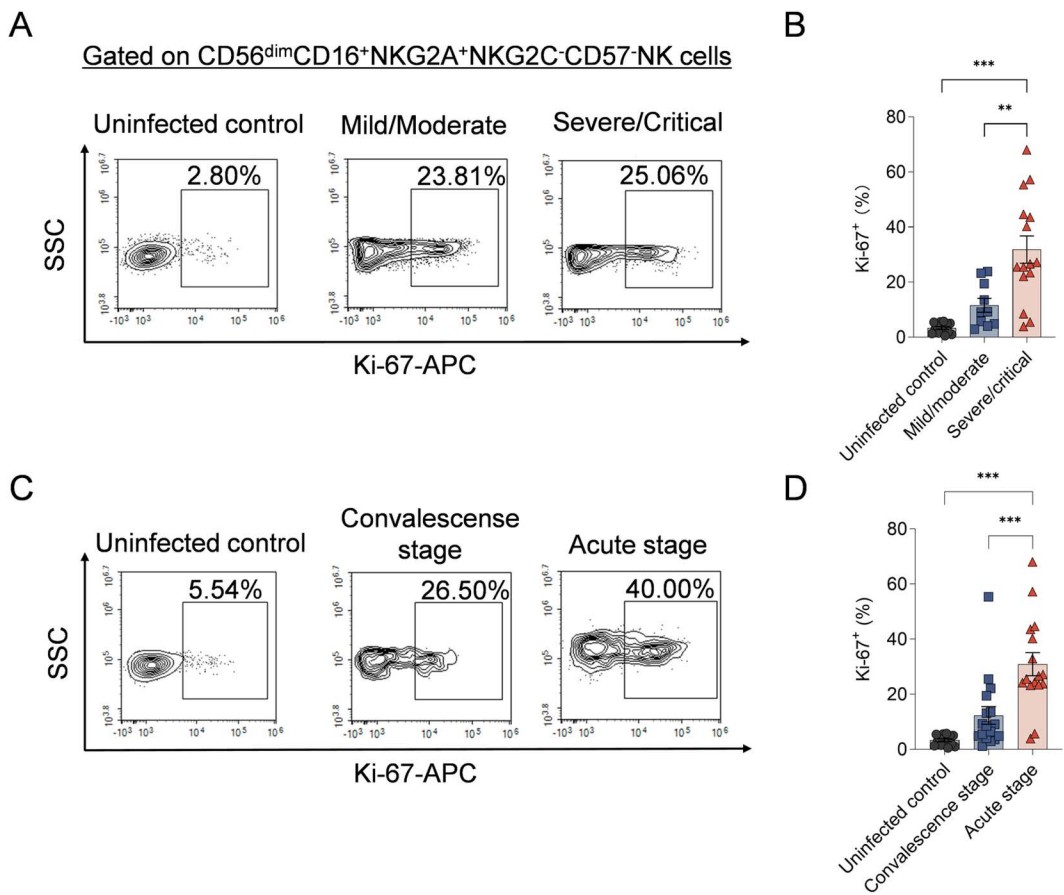

**Fig 5. The proliferation capacity of CD56^dim^CD16^+^NKG2A^+^NKG2C^-^CD57^-^ NK cells in HFRS patients with different disease severity or at different stages.** (A) Representative flow cytometric plots and (B) the comparison of the frequencies of Ki67^+^CD56^dim^CD16^+^NKG2A^+^NKG2C^-^CD57^-^ NK cells among HFRS patients with different disease severity and uninfected controls. (C) Representative flow cytometric graphs and (D) the comparison of the frequencies of Ki67^+^CD56^dim^CD16^+^NKG2A^+^NKG2C^-^CD57^-^ NK cells among HFRS patients at different disease stages and uninfected controls. Statistical analysis was performed using the Mann-Whitney $U$ test. $p$-values below 0.05 were considered statistically significant ($p<0.01$, **; $p<0.001$, ***).

We further examined the production of cytotoxic mediator perforin and granzyme B, as well as the degranulation of CD56^dim^NKG2A^+^ NK cells. Although no significant differences were observed between HFRS patients and uninfected controls in the frequencies of perforin^+^ or granzyme B^+^CD56^dim^NKG2A^+^ NK cells (Fig 7A-D and 7G-J), the expression of CD107a was significantly higher in mild/moderate and severe/critical HFRS patients compared to uninfected controls ($p<0.05$ and $p<0.01$, respectively) (Fig 7E and 7F). When comparing different disease stages, CD107a expression was elevated during the acute stage and remained at higher levels during the convalescent stage ($p<0.01$). The frequency of CD107a^+^CD56^dim^NKG2A^+^ NK cells was also higher at both stages compared to uninfected controls ($p<0.05$ and $p<0.001$, respectively) (Fig 7G and H). Moreover, we compared the expression profiles of perforin, granzyme B, and CD107a between NKG2A^+^ and NKG2A^-^ NK cells using CD56^dim^CD16^+^NKG2A^-^ NK cells as a control. Our analysis revealed no significant differences in the expression of these markers between NKG2A^+^ and NKG2A^-^ NK cells in uninfected controls (S3A-F Fig).

## Four peptides derived from HTNV bind to HLA-E*0103 stably in a concentration dependent manner

The expression of HLA-E on the surface of K562/HLA-E*0103 cells were enhanced with FI≥1 when loading with the HTNV-derived peptides NP5, GP7, GP8 and GP20 respectively. As a result, the four peptides NP5, GP7, GP8 and GP20

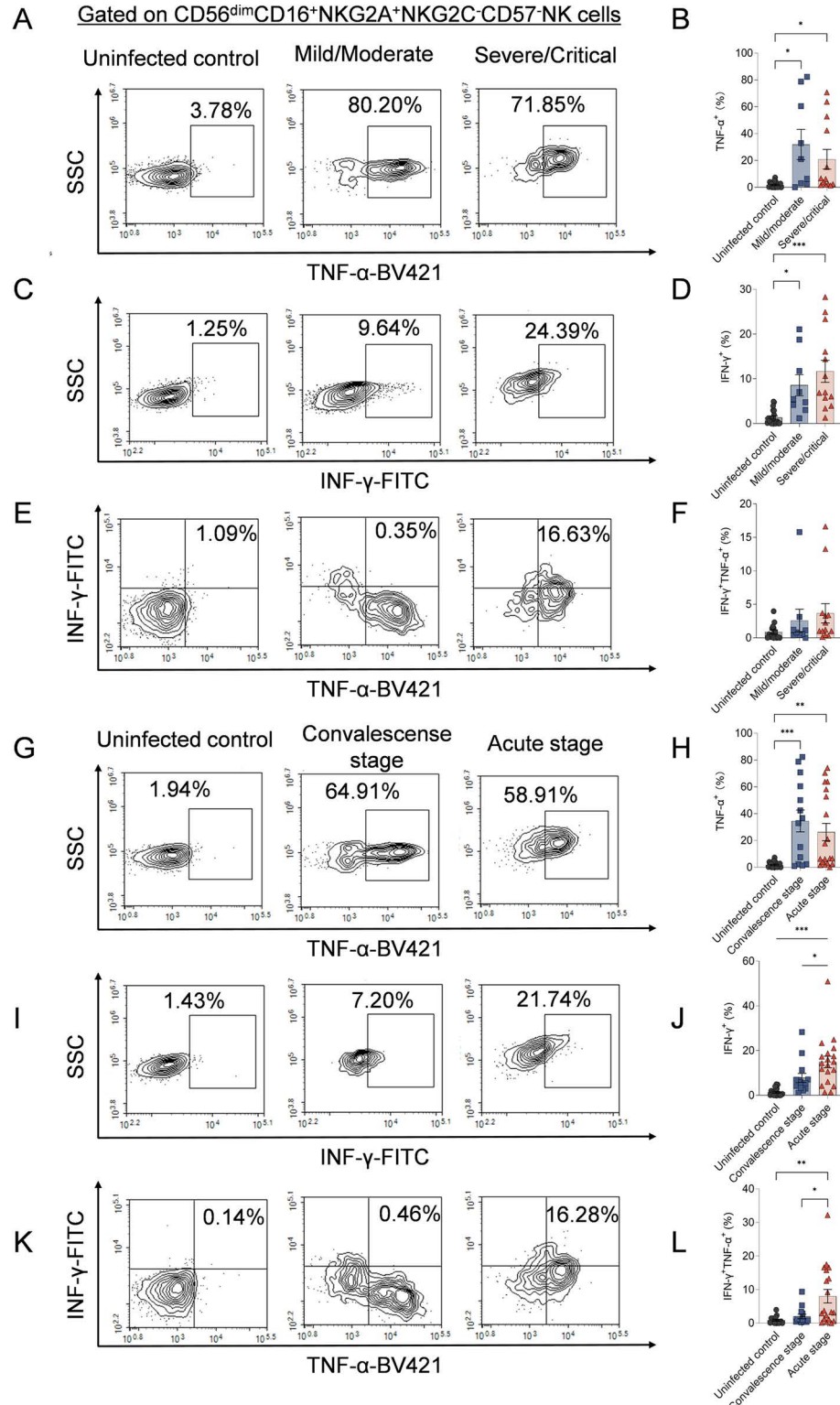

**Fig 6. The frequencies of TNF-α and IFN-γ secretion of CD56dimCD16+NKG2A+NKG2C-CD57- NK cells in HFRS patients with different disease severity or at different stages.** Representative flow cytometric plots and comparison of the frequencies of TNF-α (A and B) and IFN-γ (C and D) production- CD56dimCD16+NKG2A+NKG2C-CD57- NK cells among mild/moderate HFRS patients, severe/critical HFRS patients and uninfected controls

respectively. (E) Representative flow cytometric plots and (F) comparison of the frequencies of TNF-α⁺IFN-γ⁺CD56^dimCD16⁺NKG2A⁺NKG2C⁻CD57⁻ NK cells among mild/moderate HFRS patients, severe/critical HFRS patients and uninfected controls respectively. Representative flow cytometric plots and the comparison of the frequencies of IFN-γ (G and H) and TNF-α (I and J) production- CD56^dimCD16⁺NKG2A⁺NKG2C⁻CD57⁻ NK cells in HFRS patients at different disease stages and uninfected controls, respectively. (K) Representative flow cytometric plots and (L) comparison of the frequencies of TNF-α⁺IFN-γ⁺CD56^dimCD16⁺NKG2A⁺NKG2C⁻CD57⁻ NK cells in HFRS patients at different disease stages and uninfected controls, respectively. Statistical analysis was performed using the Mann-Whitney $U$ test. The $p$-values less than 0.05 were considered to be statistically significant ($p < 0.05$, *; $p < 0.01$, **; $p < 0.001$, ***).

could be considered as high-affinity peptides binding to HLA-E*0103 (Fig 8A). The detailed information for the screened peptides were summarized in Table 2. Incubation with a well-described HLA-E-stabilizing peptide (VMAPRTLIL) derived from the leading sequence of HLA-Cw*03 or a non-binding irrelevant peptide melanoma antigen-encoding gene (MAGE-1) (EADPTGHSY)/ human cytomegalovirus (HCMV) phosphoprotein (PP) 65 (NLVPMVATV) was used as the positive control or the negative control, showing the highest or lowest levels of HLA-E expression respectively. The results of concentration dependent assay showed that each of the four HTNV peptides was capable of stabilizing the HLA-E expression on K562/HLA-E*0103 cells at saturating concentrations ranging from 100 μM to 500 μM, which exhibited a relatively lower capacity to that of the positive control peptide (Fig 8B). The binding stability evaluation showed a declining MFI of HLA-E expression on K562/HLA-E*0103 cells when pulsing with each of the four HTNV peptides from 0 min to 480 min (Fig 8C). These results demonstrated that the HTNV-derived peptides NP5, GP7, GP8 and GP20 could bind to HLA-E*0103 and form stable HLA-E/peptide complexes, although to different degrees.

### The HLA-E/HTNV peptide complex failed to bind with CD94/NKG2A(C) receptors on CD56^dimNKG2A⁺ NK cells

To assess whether the HLA-E/HTNV peptide complex could be recognized by CD94/NKG2A(C) on NK cells, potential receptor-ligand interactions were detected by incubating recombinant CD94/NKG2A(C) heterodimers with peptide-pulsed K562/HLA-E*0103 cells. The results showed that there was few CD94/NKG2A⁺HLA-E⁺ cell after each of the four HTNV peptide incubation, whereas 47.67% of the CD94/NKG2A⁺HLA-E⁺ cells were detected after incubation with the HLA-Cw*03 leading sequence positive control peptide (Fig 9A and 9B), suggesting HLA-E/HTNV peptide complex failed to be recognized by CD94/NKG2A receptor, despite the HTNV peptides promoting the surface stabilization of HLA-E. For the recombinant CD94/NKG2C-HLA-E/HTNV peptide interaction, both HTNV peptides and positive peptide could not interact with CD94/NKG2C (Fig 9C and 9D).

Furthermore, the HLA-E*0103/peptide NP5 or GP8 tetramer staining was used to evaluate the binding capacity of HLA-E/HTNV peptide complex to CD94/NKG2A on CD56^dimNKG2A⁺ NK cells in HFRS patients (Fig 10A). The results showed that the frequency of tetramer⁺CD56^dimNKG2A⁺ NK cells in HFRS patients was 1.94% for NP5 and 0.34% for GP8, indicating the un-bound of the CD94/NKG2A and HLA-E/HTNV peptide complex. Moreover, there was no difference in the frequencies of NP5 or GP8 tetramer⁺CD56^dimNKG2A⁺ NK cells between HFRS patients and uninfected controls (Fig 10B).

### CD56^dimNKG2A⁺ NK cells activated and released TNF-α ex vivo after HTNV-NP5 peptide stimulation

An *ex vivo* co-culture assay was employed to assess the function of CD56^dimNKG2A⁺ NK cells after HTNV infection. HTNV-NP5 peptide was selected as the representative peptide pulsed with K562/HLA-E cells, which were then incubated with the isolated NK cells for further validating the function of CD56^dimNKG2A⁺ NK cells. The results showed that both the frequency of HLA-DR⁺CD56^dimNKG2A⁺ NK cells of HFRS patients were significantly higher than that of uninfected controls ($p < 0.001$), as well as Ki-67⁺CD56^dimNKG2A⁺ NK cells of HFRS patients were also higher than that of uninfected controls ($p < 0.05$) (Fig 11A-D). Functionally, CD56^dimNKG2A⁺ NK cells in HFRS patients exhibited significantly higher TNF-α and perforin production than that in uninfected controls ($p < 0.05$ and $p < 0.05$, respectively) (Fig 11F-H). While there was no

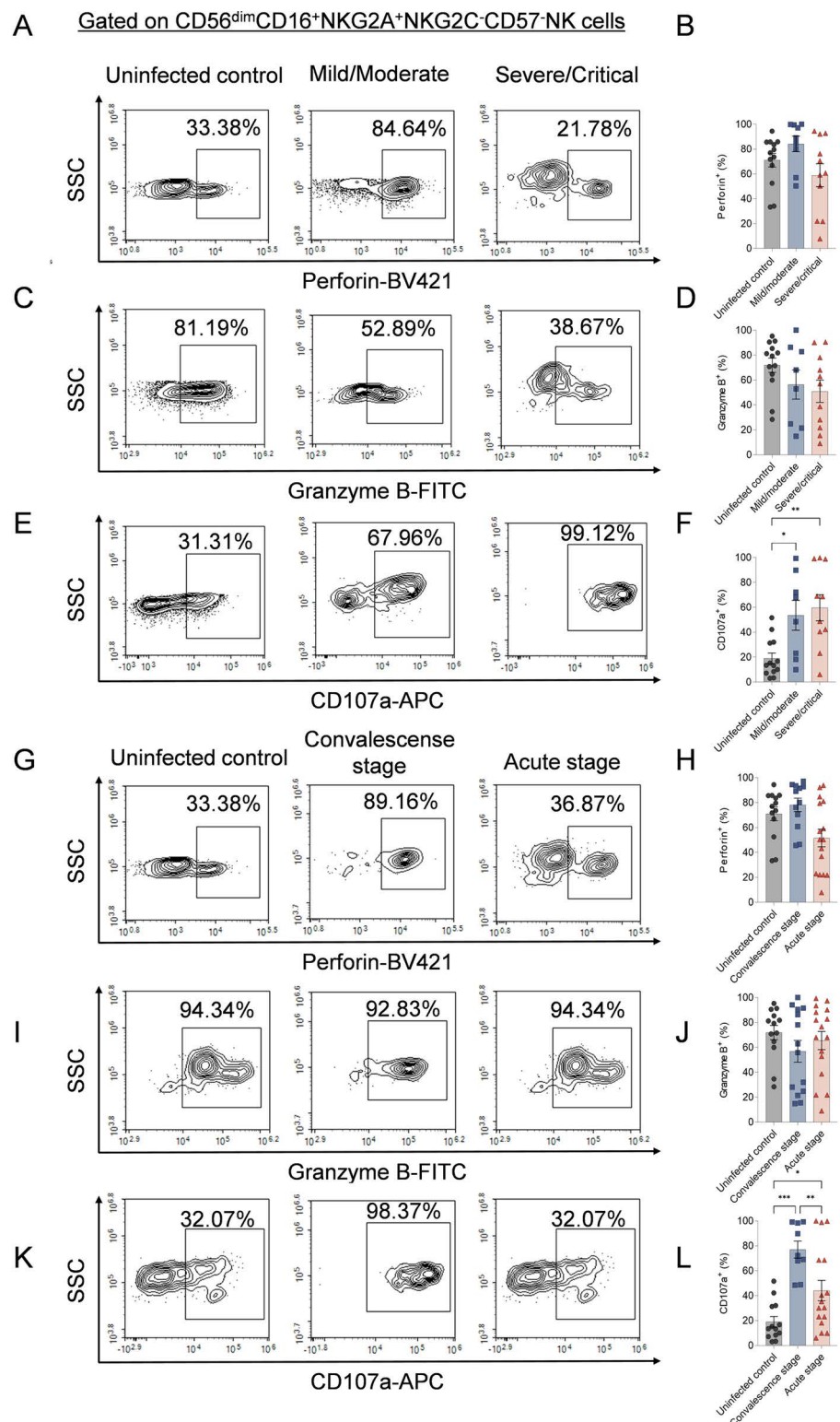

**Fig 7. The frequencies of cytotoxic mediators production of CD56$^{dim}$CD16$^{+}$NKG2A$^{+}$NKG2C$^{-}$CD57$^{-}$ NK cells in HFRS patients with different disease severity or at different stages.** Representative flow cytometric plots and the comparison of the expression of perforin (A and B), granzyme B (C and D) and CD107a (E and F) on CD56dimCD16+NKG2A+NKG2C-CD57- NK cells among mild/moderate HFRS patients, severe/critical HFRS patients

and uninfected controls, respectively. Representative flow cytometric plots and the comparison of the frequencies of perforin (G and H), granzyme B (I and J) production and CD107a (K and L) expression on CD56$^{dim}$CD16$^+$NKG2A$^+$NKG2C$^-$CD57$^-$ NK cells among patients at different stages of the illness and uninfected controls, respectively. The Mann-Whitney $U$ test was used for Statistical analysis. The $p$-values less than 0.05 were considered to be statistically significant ($p < 0.05$, *; $p < 0.01$, **; $p < 0.001$, ***).

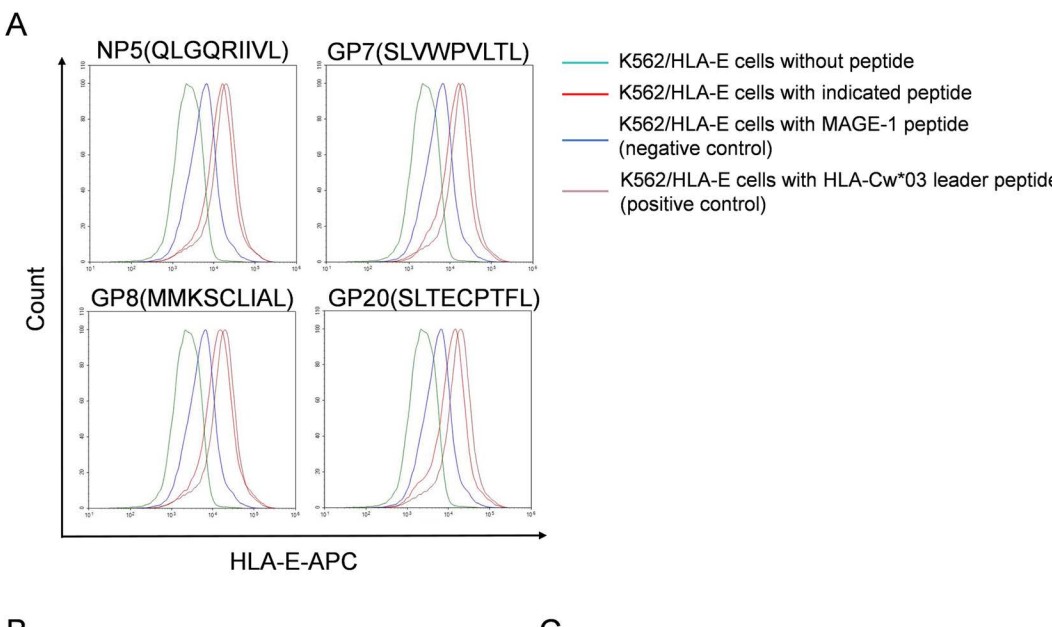

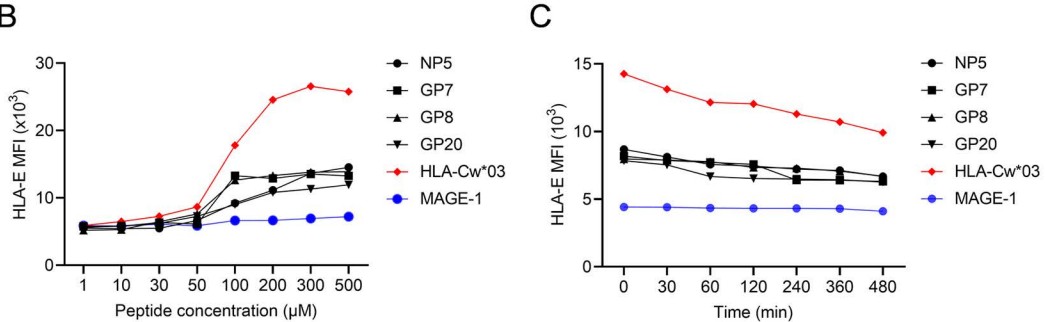

**Fig 8. The screen for the HTNV-derived peptides binding to HLA-E*0103 and evaluation of the binding stability and concentration dependence.** (A) Overlay histogram showed a rightward shift with higher fluorescence intensity (FI) in the curve of K562/HLA-E*0103 cells incubated with each of the four indicated HTNV-derived peptides compared to cells incubated without peptide, indicating binding to HLA-E*0103. (B) The mean fluorescence intensity (MFI) of HLA-E expression on K562/HLA-E*0103 cells pulsed with different peptide concentrations for each of the four indicated HTNV-derived peptides or control peptides (HLA-Cw*03 or MAGE-1). (C) The MFI change tendency of HLA-E expression at the different time points on K562/HLA-E*0103 cells pulsed with each of the four indicated HTNV-derived peptides or control peptides (HLA-Cw*03 or MAGE-1) at 100uM.

difference in the frequency of CD107a$^+$CD56$^{dim}$NKG2A$^+$ NK cells between HFRS patient group and uninfected control group after NP5 peptide stimulation (Fig 11I and J).

To further elucidate the impact of peptides presented by HLA-E on the function of CD56$^{dim}$NKG2A$^+$NK cells, we analyzed the functional phenotypes and indicators across different peptide groups. Our results demonstrated that CD56$^{dim}$NKG2A$^+$NK cells from uninfected controls, when co-cultured with K562/HLA-E cells loaded with HTNV peptides, exhibited significantly higher levels of CD107a expression compared to those co-cultured with cells presenting the HLA-Cw*03 leader peptide control ($p < 0.05$ and $p < 0.01$, respectively) (Fig 12A and 12B). Moreover, we compared the degranulation

**Table 2. Screen of HTNV-derived HLA-E peptides by the K562/HLA-E*0103 cell binding assay.**

| NO. peptide | Protein | Sequence | Location | FI |
|---|---|---|---|---|
| NP5 | HTNV-NP | QLGQRIIVL | aa375-aa383 | 2.20 |
| GP7 | HTNV-GP | SLVWPVLTL | aa11-aa19 | 1.36 |
| GP8 | HTNV-GP | MMKSCLIAL | aa153-aa161 | 1.96 |
| GP20 | HTNV-GP | SLTECPTFL | aa996-aa1004 | 2.02 |
| HLA-Cw*03 leading seq (positive control) | – | VMAPRTLVL | – | 2.55 |
| MAGE-1/HCMV PP65 (negative control) | – | EADPTGHSY/ NLVPMVATV | – | 0.05 |

FI, determined as follows: FI = (mean APC fluorescence with the given peptide - mean APC fluorescence without peptide)/ (mean APC fluorescence without peptide). FI ≥ 1 represents that the peptide could bind to HLA-E.

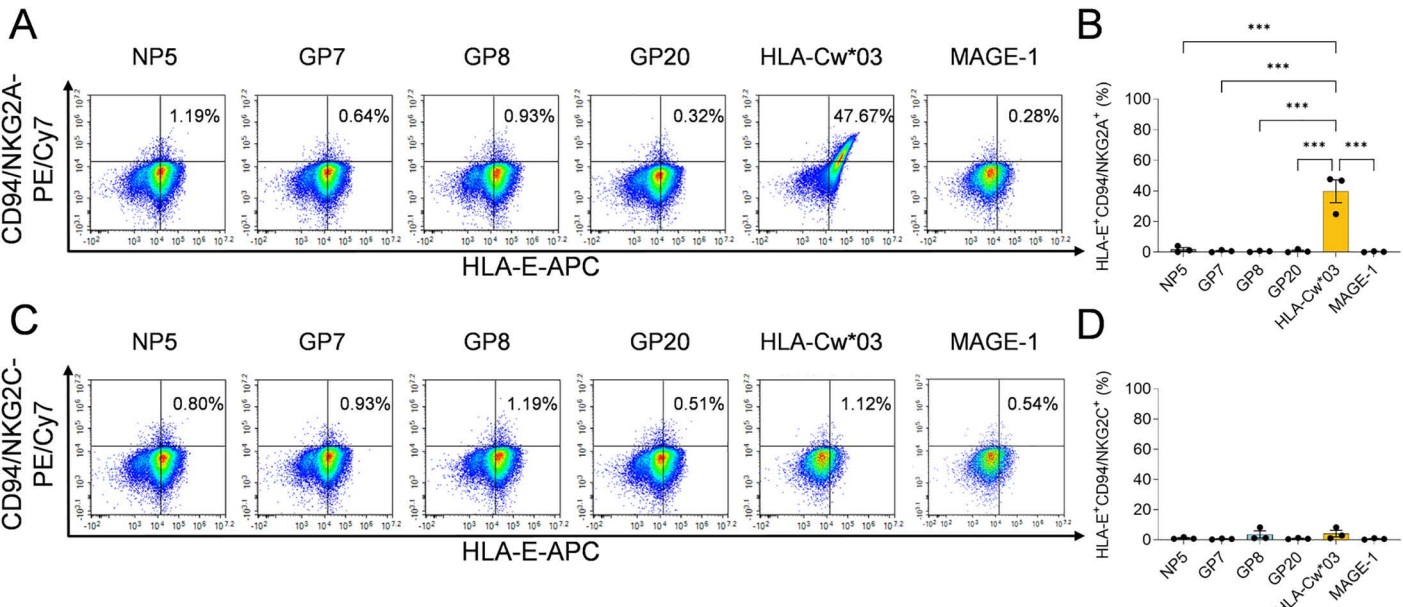

**Fig 9. Detection of the binding capacity between HLA-E/HTNV peptide complexes and CD94/NKG2A(C) receptor.** (A) Representative flow cytometric plots and (B) the comparison of the frequencies of CD94/NKG2A⁺HLA-E⁺ cells after incubation with each of the four HTNV-derived peptide or control peptide. (C) Representative flow cytometric plots and (D) the comparison of the frequencies of CD94/NKG2C⁺HLA-E⁺ cells after each of the four HTNV-derived peptide or control peptide incubation. The peptide from HLA-Cw*03 leading sequence (VMAPRTLIL) served as the positive control. The peptide from MAGE-1 (EADPTGHSY) served as the negative control. Statistical analysis was performed using the Mann-Whitney $U$ test. $p$-values below 0.05 were considered statistically significant ($p < 0.05$, *; $p < 0.01$, **; $p < 0.001$, ***).

activity between CD56ᵈⁱᵐNKG2A⁺ and CD56ᵈⁱᵐNKG2A⁻NK cells. In contrast, CD56ᵈⁱᵐNKG2A⁻ NK cells did not exhibit significant differences in degranulation across the various groups (Fig 12C and 12D). Additionally, the expression of perforin and TNF-α in both CD56ᵈⁱᵐNKG2A⁺ and CD56ᵈⁱᵐNKG2A⁻ NK cells remained unchanged across all groups (Fig 12E-L).

**The HLA-E/HTNV-derived peptide complex enhanced the cytotoxic capacity of CD56ᵈⁱᵐNKG2A⁺NK cells at the acute stage of HFRS**

To evaluate the specific cytotoxic capacity of CD56ᵈⁱᵐNKG2A⁺ NK cells against HTNV peptide-loaded targets, we conducted comparative cytotoxicity assays between CD56ᵈⁱᵐNKG2A⁺ and CD56ᵈⁱᵐNKG2A⁻NK cell subsets (Fig 13A).

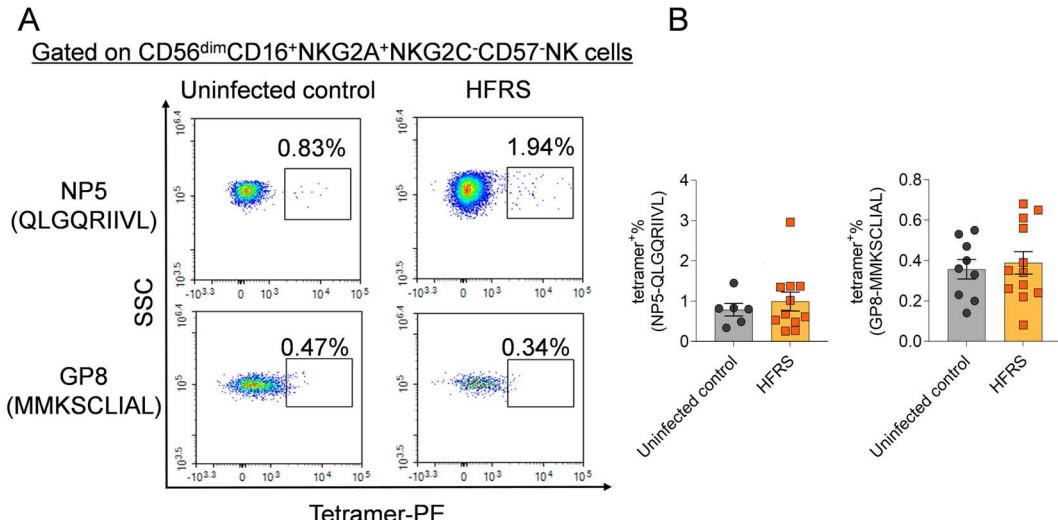

**Fig 10. The tetramer staining to detect the binding capacity of HLA-E/HTNV peptide complex to CD56ᵈⁱᵐCD16⁺NKG2A⁺ NK cells in PBMCs from HFRS patients.** (A) Representative flow cytometric plots and (B) comparison of the frequencies of HLA-E/peptide (NP5 or GP8) tetramer⁺ cells gated on CD56ᵈⁱᵐCD16⁺NKG2A⁺NKG2C⁻CD57⁻ NK cells between HFRS patients and uninfected controls.

Strikingly, at a 3:1 effector-to-target (E:T) ratio, CD56ᵈⁱᵐNKG2A⁺NK cells exhibited significantly enhanced killing activity in the GP8 peptide-stimulated group compared to the HLA-Cw*03 leader peptide control ($p<0.05$), as evidenced by elevated percentages of EGFP⁺Fixable Viability Dye eFluor 450⁺ dead cells (Fig 13B). This GP8-specific enhancement, however, was not observed in CD56ᵈⁱᵐNKG2A⁻ NK cells. Importantly, neither subset demonstrated differential cytotoxicity between the HLA-Cw*03 leader peptide control and groups lacking peptide stimulation, exposed to alternative control peptides, or treated with other HTNV-derived peptides (Fig 13B and 13C), highlighting the selective response to GP8 peptide in CD56ᵈⁱᵐNKG2A⁺ NK cells.

## Discussion

NK cells participate in host innate immune response against virus infection through direct cytotoxicity as well as cytokine secretion [22]. The balance between activation and inhibitory receptors on NK cells is closely related to the functional effects of different NK cell subsets. In this study, CD56ᵈⁱᵐ NK cell subset in the peripheral blood of HFRS patients showed a significant upregulated expression of NKG2A. The CD56ᵈⁱᵐNKG2A⁺ NK cells of HFRS patients displayed a highly activation phenotype and proliferation capacity, as along with increased secretion of cytokines and cytotoxic mediators. Moreover, HTNV-derived peptides that could be presented by HLA-E were identified. However, HLA-E/HTNV peptide complex could not interact with CD94/NKG2A on CD56ᵈⁱᵐ NK cells. The CD56ᵈⁱᵐNKG2A⁺ NK cells of HFRS patients showed enhanced cytotoxic activity *ex vivo* against HTNV peptide-loaded K562/HLA-E cells. These findings may provide a possible mechanism for the CD56ᵈⁱᵐNKG2A⁺ NK cell subset to response against HTNV infection through the abrogation of inhibitory signals for NK cell activation. The HLA-E/HTNV-derived peptide-NKG2A axis-mediated NK cell immune response may be involved in the pathogenesis of HFRS disease.

Previous study on PUUV-infected HFRS patients found that the number of CD56⁺CD16⁺ NK cells in bronchoalveolar lavage (BAL) fluid of patients was increased, whereas the number of NK cells in the peripheral blood of the PUUV-infected patients decreased at the acute stage of HFRS compared with the healthy individuals [23,24]. In the peripheral blood of PUUV-infected patients, the CD56ᵈⁱᵐ NK cells rapidly expanded approximately around day 10 after the onset of clinical symptoms and maintained proliferation for at least 2 months [10]. Furthermore, in DOBV-infected patients, a higher

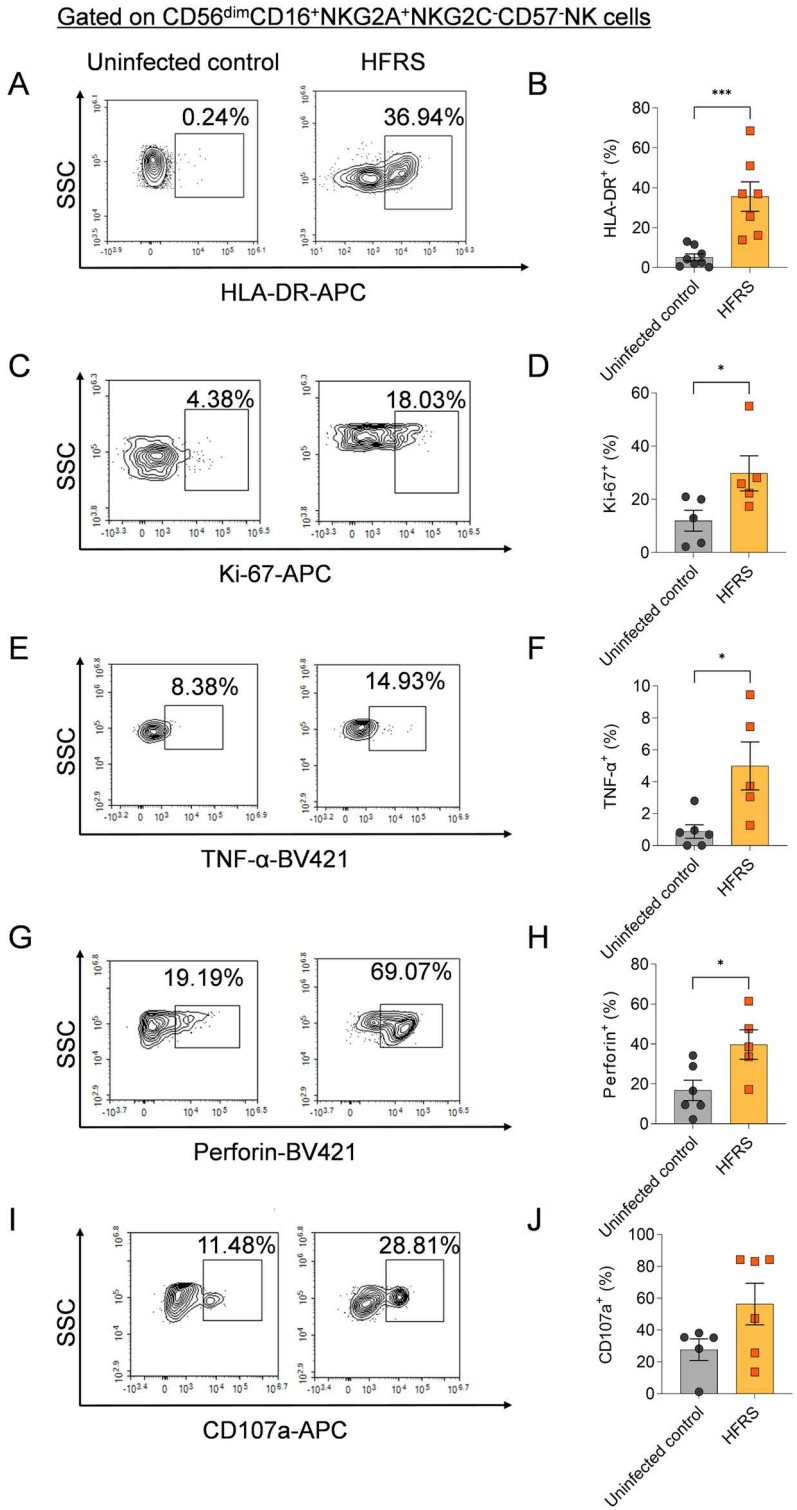

Gated on CD56$^{dim}$CD16$^+$NKG2A$^+$NKG2C$^-$CD57$^-$NK cells

**Fig 11. Activation, proliferation and effector capacities of CD56$^{dim}$NKG2A$^+$NK cells in HFRS patients by co-culture with HTNV-NP5 peptide-pulsed K562/HLA-E\*0103 cells *ex vivo*.** Representative flow cytometric plots and changes in the frequencies of activated (HLA-DR$^+$) (A and B), proliferated (Ki67$^+$) (C and D) and functional (TNF-α$^+$, perforin$^+$ and CD107a$^+$) (E-J) CD56$^{dim}$NKG2A$^+$NK cells between HFRS patients and uninfected controls in the presence of the HTNV-NP5 peptide. Total CD3$^-$CD56$^+$NK cells were isolated from the PBMCs of each HFRS patient and uninfected control using a human NK cell

negative selection kit. The isolated NK cells were then co-cultured with HTNV-NP5 peptide-pulsed K562/HLA-E*0103 cells *ex vivo.* Statistical analysis was performed using the Mann-Whitney *U* test, with p-values below 0.05 considered statistically significant ($p < 0.05$, *; $p < 0.01$, **).

percentage of CD3<sup>-</sup>CD56<sup>+</sup> NK cells were also observed [25]. In HTNV-infected HFRS patients, an increased number of NK cells was observed in peripheral blood of the patients, particularly in those with mild symptoms [2]. The changes of number or frequency for NK cells suggested that NK cell responses may play an important role in hantavirus infection. Based on these findings, the subclusters and transcriptional signatures of NK cells in peripheral blood of HTNV-infected HFRS patients was firstly analyzed in scRNA-seq in this study. The proportion of the NK cell cluster 4 and cluster 5 in the PBMCs of the HFRS patients were increased, which was further confirmed to display CD56<sup>dim</sup>CD16<sup>+</sup>NKG2A<sup>+</sup>NKG2C<sup>-</sup>CD57<sup>-</sup> phenotype in HFRS patients. It has been reported that NK cells in human peripheral blood could be subclustered as immature (CD56<sup>bright</sup>CD16<sup>-</sup>), mature (CD56<sup>dim</sup>CD16<sup>+</sup>CD57<sup>-</sup>) and terminally differentiated (CD56<sup>dim</sup>CD16<sup>+</sup>CD57<sup>+</sup>) subsets according to the scRNA-seq analysis [26,27]. Therefore, the elevated subcluster of NK cells in HFRS patients may be the mature NK cells with the immune effects closely related with NKG2A.

Further analysis found that the expression of genes characterizing inflammation and activation, along with the enriched TNF signaling pathway were upregulated in CD56<sup>dim</sup>NKG2A<sup>+</sup>NK cells of HFRS patients with mild or moderate severity. Whereas high expression of proliferation makers were the main characteristics for CD56<sup>dim</sup>NKG2A<sup>+</sup>NK cells in severe or critical HFRS patients. The significant difference in characteristic genes expression among HFRS patients with different severity suggested that the function of CD56<sup>dim</sup>NKG2A<sup>+</sup>NK cell subset may be involved in the pathogenesis of HTNV infection.

Epidemiological studies of HFRS have consistently reported a significant gender disparity in the incidence of the disease. For example, a study conducted in Zhejiang Province found that the male-to-female ratio among HFRS patients across all age groups was 2.6:1 [28]. Similarly, data from Jilin Province between 2018 and 2023 revealed a pronounced predominance of male patients, with a total of 1,511 male cases compared to 482 female cases, resulting in a gender ratio of 3.13:1 [29]. The sample size of HFRS patients, though aligned with recent epidemiological trends in China, remains relatively small, which is an obvious limitation for this study. While our findings suggest associations between NK cell characteristics and disease severity, these observations should be interpreted as preliminary and require future validation in larger, multi-center cohorts.

The increased frequency of CD56<sup>dim</sup>CD16<sup>+</sup>NKG2A<sup>+</sup>NKG2C<sup>-</sup>CD57<sup>-</sup> NK cell subset in peripheral blood of HFRS patients was further confirmed in *ex vivo* FCM detection, which was in line with the analysis of scRNA-seq. The overexpression of NKG2A on NK cells was observed in several virus infected diseases. In hepatitis B virus (HBV) infection, the overexpression of NKG2A on NK cells lead to severe impairment of NK cells cytotoxicity in CHB patients [17,30]. In human cytomegalovirus (HCMV) infected patients, the NKG2C<sup>+</sup> NK cell was the main subset undergoing clonal-like expansion that developed anti-virus responses [31]. During SARS-CoV-2 infection, exhausted NK cells showed the upregulation of NKG2A, and the expression of NKG2A on NK cells was markedly reduced at convalescence stage of the patients [32]. An increased frequency of NKG2A<sup>+</sup>CD57<sup>+</sup>CD56<sup>dim</sup> NK cells (adaptive NK cells) were observed in severe SARS-CoV-2 infected patients [33]. Therefore, the regulation of different NK cell subsets with NKG2A or NKG2C expression during virus infection may have a significant impact on the function of NK cells, which is closely related to the antiviral activity or the ability of the virus to evade the immune response. Importantly, our findings showed that both the MFI and frequency of NKG2A on CD56<sup>dim</sup>CD16<sup>+</sup> NK cells of HFRS patients were significantly higher than that of NKG2C. The function and the underling mechanisms of the CD56<sup>dim</sup>CD16<sup>+</sup>NKG2A<sup>+</sup> NK cell subset were then explored in this study.

Several studies have shown that HLA-DR can be used as a late activation marker for NK cells in various infectious diseases, with activated HLA-DR<sup>+</sup>NK cells peaking within 3 to 4 weeks after symptom onset and then declining. Additionally, CD69 is an early activation marker of NK cells, and CD69<sup>+</sup>CD56<sup>dim</sup>NK cells are significantly elevated in severe dengue patients even during recovery. In our study, we detected CD69 expression on CD56<sup>dim</sup>NKG2A<sup>+</sup>NK cells in PBMCs from 19 acute-phase and

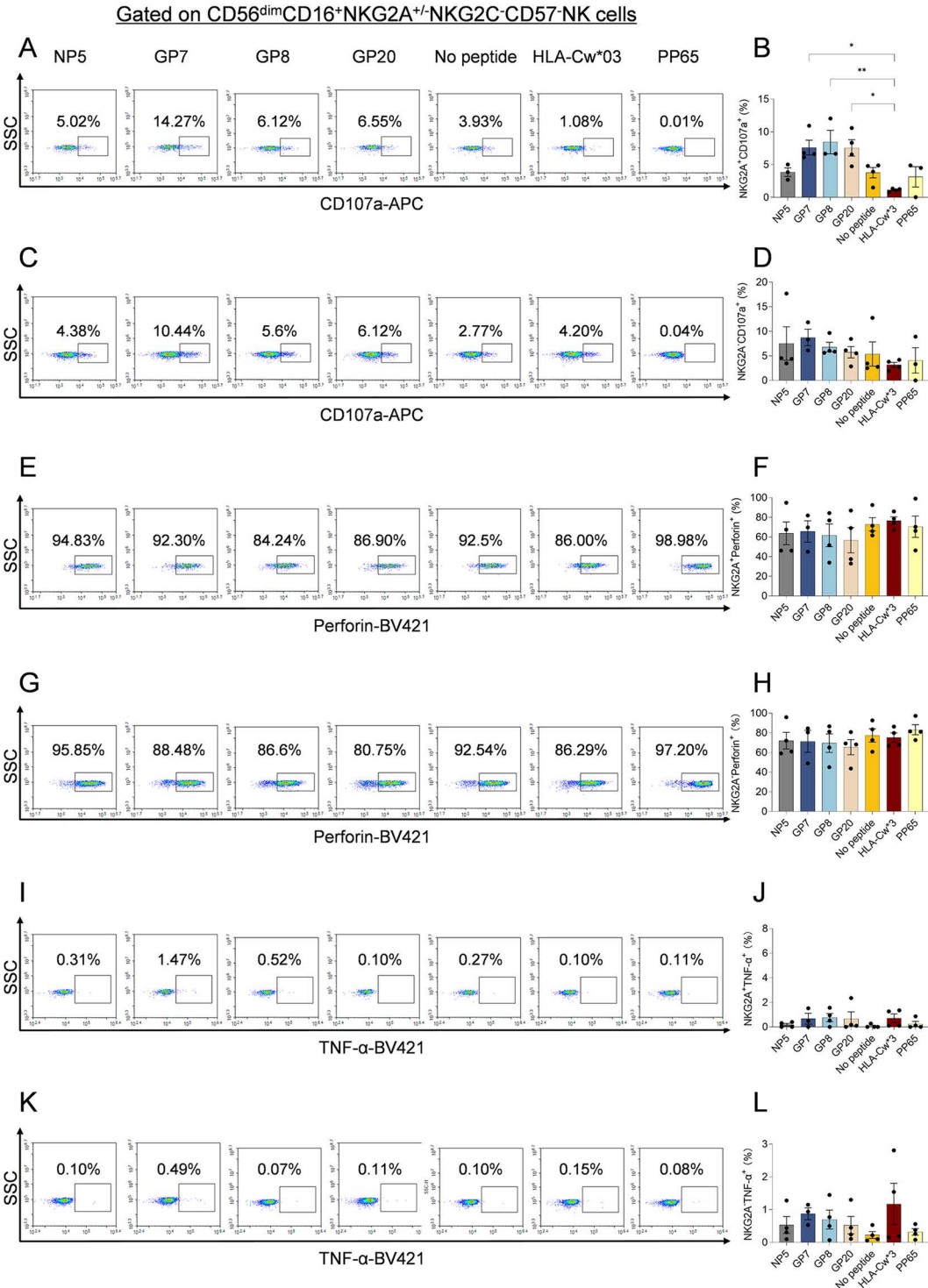

Gated on CD56$^{dim}$CD16$^+$NKG2A$^{+/-}$NKG2C$^-$CD57$^-$NK cells

**Fig 12. Activation, proliferation, TNF-α and cytotoxic mediator production of CD56$^{dim}$NKG2A$^+$NK cells upon co-culture with K562/HLA-E cells pulsed with the indicated peptides *ex vivo*.** Representative flow cytometric (FCM) analysis and cumulative results showing the frequencies of CD107a$^+$ (A-D), perforin$^+$ (E-H) and TNF-α$^+$ (I-L) in CD56$^{dim}$NKG2A$^+$ and CD56$^{dim}$NKG2A$^-$NK cells from PBMCs of uninfected controls across different peptide groups. K562/HLA-E*0103 cells loaded with peptide from HLA-Cw*03 leading sequence (VMAPRTLIL) were used as the positive control, while K562/HLA-E*0103 cells loaded with the peptide from HCMV phosphoprotein PP65 (NLVPMVATV) or no peptide were used as negative controls. Statistical analysis was performed using the Mann-Whitney *U* test, with p-values below 0.05 considered statistically significant ($p < 0.05$, *; $p < 0.01$, **).

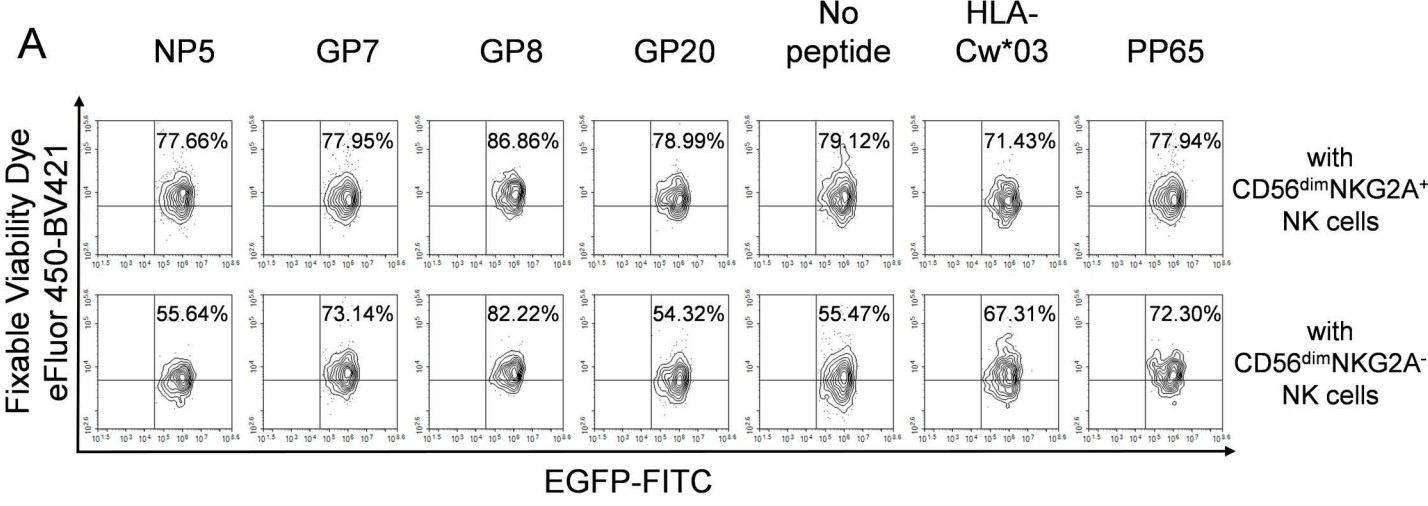

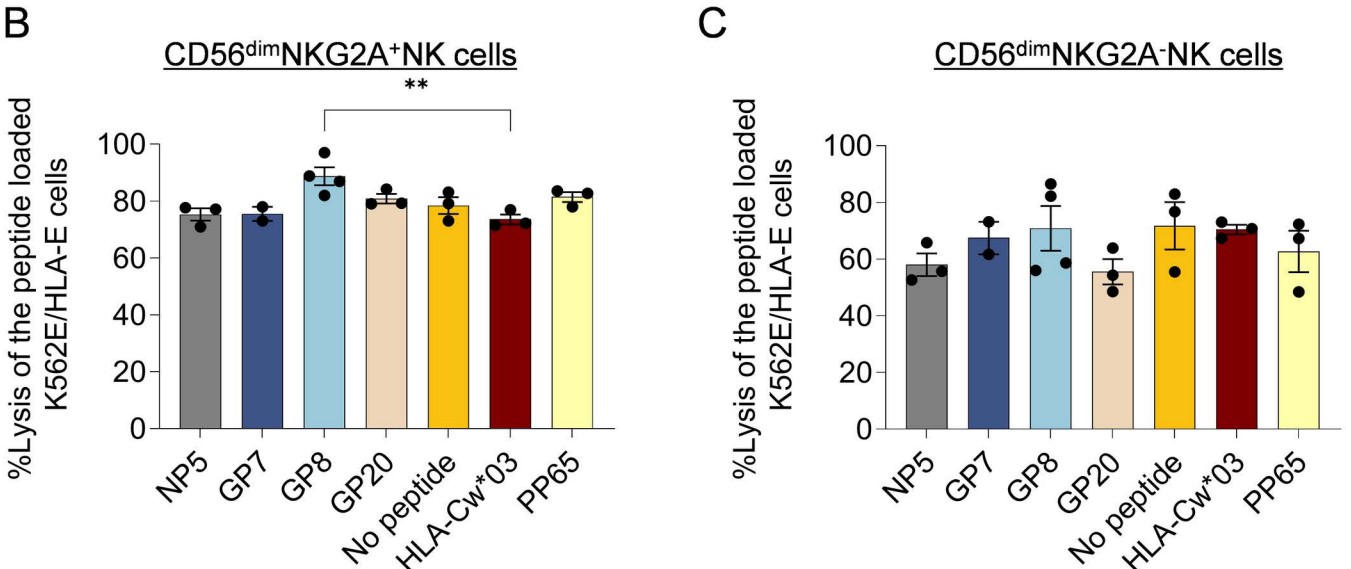

**Fig 13. Ability of the HLA-E/HTNV-derived peptide complex to induce CD56$^{dim}$NKG2A$^+$NK cell-mediated lysis.** (A) Representative flow cytometric plots and (B) Kinetics and comparison showing the percentages of targeted cells killed by CD56$^{dim}$NKG2A$^+$ and CD56$^{dim}$NKG2A$^-$ NK cells at effector-to-target cell ratios of 3:1 in different peptides groups. EGFP-labeled K562/HLA-E*0103 cells loaded with indicated peptide were used as target cells, while FACS-sorted CD3$^-$CD56$^{dim}$NKG2A$^+$ and CD3$^-$CD56$^{dim}$NKG2A$^-$ NK cells from the PBMCs of uninfected controls were used as effector cells. K562/HLA-E*0103 cells loaded with the peptide from the HLA-Cw*03 leading sequence (VMAPRTLIL) served as the positive control, whereas K562/HLA-E0103 cells loaded with the peptide from HCMV phosphoprotein PP65 (NLVPMVATV) or no peptide served as negative controls. Fixable Viability Dye eFluor 450 was used to label dead cells.

7 convalescent-phase patients, aiming to further investigate the activation status of NK cells in the context of specific infections. Moreover, the correlation between higher frequencies of CD69$^+$CD56$^{dim}$NKG2A$^+$ NK cells and the milder HFRS disease further confirmed the function of the CD56$^{dim}$NKG2A$^+$ NK cell subset may partially determine the prognosis of HFRS patients.

Upon SARS-CoV-2 infection, an increased expression of the proliferation marker Ki67 was observed in NK cells with anti-virus functions [34]. The upregulation of Ki67 in NK cells is a sign of their activation and proliferation to combat the virus. The scRNA-seq analysis in our study showed that the NK cell cluster 5 exhibited significant expression

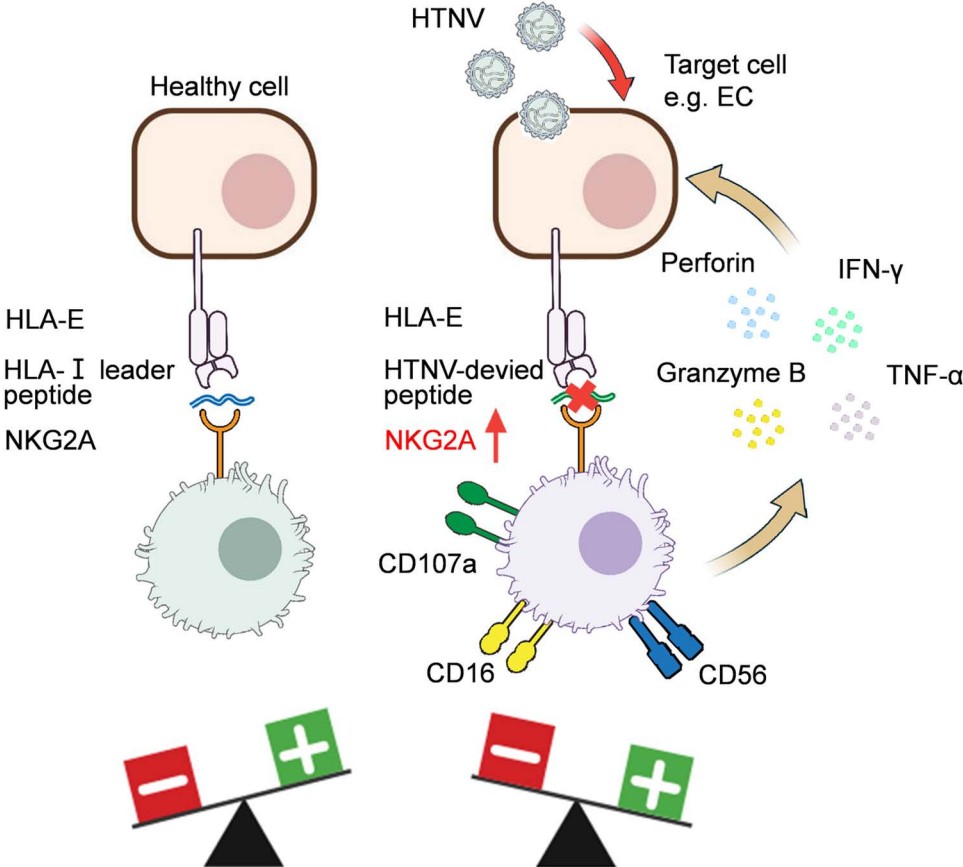

**Fig 14. Proposed model illustrating the interaction between HLA-E/peptide complexes and NKG2A on NK cells.** Normally, HLA-E presented peptides derived from the leading sequence of classical HLA class I (HLA-Ia) and HLA-G molecules. Therefore, NKG2A could inhibit the cytotoxicity of NK cells to autologous healthy cells through interacting with the HLA-E/peptide complex (left). During HTNV infection, HLA-E formed a complex with HTNV-derived peptides. The HLA-E/HTNV peptide complex was unable to interact with NKG2A, leading to the enhanced antiviral activity of NK cells (right).

of MKI67 gene in critical HFRS patients, which was partially confirmed by FCM staining with PBMCs of HFRS patients. The CD56$^{dim}$NKG2A$^+$NK cells indeed showed enhanced proliferation capacity at the acute stage of HFRS. However, the stronger proliferation of CD56$^{dim}$NKG2A$^+$NK cells was associated with more severe HFRS, indicating NK cells might exert antiviral effects while also participating in the pathogenesis of HFRS. There might be a complex interplay between different subsets of NK cell immune responses and disease manifestations.

When the immune system is combating a viral infection, NK cells may exert a dual role, for they not only produce TNF-α and IFN-γ to modulate the immune response but also directly kill virus-infected cells through cytotoxic mediators such as perforin and granzymes. The balance between these activities might be crucial in determining the outcome of the infection [30]. In mild HFRS patients, CD56$^{dim}$NKG2A$^+$NK cells could synthesize significant high level of TNF-α, suggesting the inflammation response mediated by CD56$^{dim}$NKG2A$^+$NK cells may be beneficial for virus control. While in severe HFRS patients, CD56$^{dim}$NKG2A$^+$NK cells could enhance the production of IFN-γ. Similarly, in severe SARS-CoV-2 infected patients, NK cells were functionally impaired and enrich IFN signaling, suggesting IFN might induce NK cells response with poorer disease outcome [35,36]. Moreover, we found that the CD56$^{dim}$NKG2A$^+$NK cells of HFRS patients could produce perforin and express high level of degranulation marker CD107a, while produce similar level of granzyme

B compared with uninfected controls in this study, which was in accordance with the previous report that CD56[dim]NK-G2A[+]CD57[-]NK cell subset could highly expressed granzyme B in healthy individuals [27]. Similar with our findings, the levels of perforin were elevated in CD56[dim] NK cells in PUUV-infected patients [10], as well as the frequency of CD107a[+]CD56[dim] NK cells was increased during HTNV infection [37]. Therefore, our data suggested that CD56[dim]NK-G2A[+]NK cells may perform anti-virus activity through direct cytotoxicity as well as TNF-α and IFN-γ secretion. These pre-liminary findings suggest a potential link between NK cell functional profiles and HFRS severity, though further validation in expanded cohorts is necessary to establish causality.

Based on the fact that the interaction between NKG2A and HLA-E/peptides could regulate activity of NK cells, we further speculated whether there were HTNV-derived peptides presented by HLA-E that affected the function of CD56[dim]NKG2A[+] NK cells. Our previous study has identified the HTNV 9-mer epitopes restricted by HLA-E which could induce protective CD8[+] T cell responses in HFRS patients [38]. The HLA-E restricted effective specific CD8[+] T cell responses must be induced by TCR-HLA-E/HTNV epitope interaction [38]. Meanwhile, HLA-E is the ligand for CD94/NKG2A or CD94/NKG2C expressed on NK cells and activated CD8[+] T cells. Given to the multi-receptors for HLA-E as well as the corresponding activation or inhibitory signals transduction, it seems that HLA-E may play an important role in balancing the immune responses to HTNV infection. Several studies have found that the cytotoxicity of NKG2A[+] NK cell could be inhibit if the HLA-E/viruses peptide complex could be recognized and interacted with CD94/NKG2A [12]. However, when the virus-derived peptide presented by HLA-E could not be interacted with CD94/NKG2A on NK cells, the inhibition for NK cells through HLA-E-NKG2A axis would be abrogated. A representative example is SARS-COV-2 infection, in which NK cells could receive activated signal or inhibitory signal through the interaction between CD94/NKG2A and HLA-E loaded with different peptides. CD94/NKG2A on NK cells could recognize and interact with HLA-E/peptide (LQPRTFLL) complex derived from SARS-CoV-2 spike 1 protein (SP1), leading to the inhibitory signal for NK cells and the exhaustion of NK cells at the early stages of COVID-19 [39]. While SARS-CoV-2 non-structural protein (nsp) 13 derived peptide (VMPLSAPTL) presented by HLA-E could not be interacted with CD94/NKG2A on NK cells, thereby generating the abrogation of inhibitory signals and contributing to the elevated NK cell antiviral activity in a "missing-self" mechanism [20].

Four HTNV peptides that stabilize HLA-E were identified, including one on the nucleocapsid of HTNV and three on the glycoprotein of HTNV. The dose-dependent manner and binding stability of the HTNV-derived peptides in stabilizing the expression of HLA-E was similar to the HLA-Cw*3 leading sequence peptide. However, the binding affinity of each HTNV peptide to HLA-E was lower than that of the leading sequence peptide, especially under the 100 μM to 500 μM incubation condition. Therefore, the possible mechanism of HTNV-derived peptide substituting for the HLA-I leading sequence to bind with HLA-E requires further investigation. Both the receptor-ligand binding assay and the peptide/HLA-E*0103 tetramer staining revealed that there was almost no interaction between CD94/NKG2A and HLA-E/HTNV-derived pep-tides. Each of the four HLA-E/HTNV-derived peptide complexes failed to interact with the CD94/NKG2A receptor, even though the peptides could promote the stable expression of HLA-E on the surface of target cells. Consistent with the study on human immunodeficiency virus (HIV), a HIV-derived peptide AISPRTLNA (AA9) presented by HLA-E on HIV-infected cells was also unable to be recognized by NKG2A/CD94 on NK cells, resulting in the lysis of HIV-infected cells by NK cells expressing NKG2A/CD94 [40]. Our data demonstrated that the HLA-E/HTNV-derived peptide complexes failed to bind with the CD94/NKG2A receptor, rendering HTNV-infected cells highly susceptible to attack by CD56[dim]NKG2A[+] NK cells.

The *ex vivo* co-culture of purified NK cells from the PBMCs of HFRS patients and HTNV peptide pulsed K562/HLA-E*0103 cells was then conducted to find out the direct evidence that interaction between CD94/NKG2A and HLA-E/HTNV peptide complex would affect the antiviral ability of CD56[dim]NKG2A[+] NK cells. Indeed, CD56[dim]NKG2A[+] NK cells from the peripheral blood of HFRS patients displayed stronger activation, higher levels of TNF-α production and degranulation ability under the treatment of HLA-E/HTNV-NP5 peptide, especially when compared with CD56[dim]NKG2A[-] NK cells. Furthermore, our findings collectively demonstrate that HLA-E-restricted presentation of the HTNV-GP8 peptide (MMKSCLIAL) selectively enhances the cytotoxic potential of CD56[dim]NKG2A[+] NK cells by evading CD94/NKG2A-mediated inhibitory

signaling. Notably, *in vitro* co-culture assays revealed that the HLA-E/HTNV-GP8 peptide complex significantly boosted the killing capacity of CD56$^{dim}$NKG2A$^+$NK cells against HLA-E-expressing targets (Fig 13), a phenomenon not observed with control peptides such as the HLA-Cw*03 leader sequence (VMAPRTLIL) that actively engage the CD94/NKG2A axis. Specifically, the structural incompatibility between HTNV-derived peptides (including GP8) and the CD94/NKG2A receptor likely prevents inhibitory signaling, thereby "unshackling" CD56$^{dim}$NKG2A$^+$ NK cells to execute anti-viral cytotoxicity. These insights highlight the therapeutic potential of designing HLA-E-restricted peptides that selectively modulate NK cell responses against viral infections. Notably, the HTNV-NP5 peptide exhibited strong HLA-E*0103 binding (FI=2.20) yet failed to engage CD94/NKG2A-mediated NK cell inhibition (Fig 13B and 13C). This functional-structural dissociation suggests NP5 may potentially compete with inhibitory peptides to enhance NK cell surveillance while retaining immunogenicity for CD8$^+$ T cell activation. Structural characterization of the HLA-E/NP5/NKG2A interface is warranted to resolve this mechanism in future.

Our findings in this study distinctly indicate that NK cells in HFRS patients predominantly possess the phenotypes of CD56$^{dim}$CD16$^+$NKG2A$^+$NKG2C$^-$CD57$^-$, demonstrating remarkable abilities in activation, proliferation, secretion of TNF-α, IFN-γ and cytotoxic mediators. HTNV-derived peptides could be presented by HLA-E on HTNV-infected cells, but they prevent the recognition and interaction between HLA-E and CD94/NKG2A on CD56$^{dim}$NKG2A$^+$NK cells in the peripheral blood of HFRS patients, thus "abrogating" the inhibitory signal of NKG2A and "relieving" the cytotoxic effects of NK cells against HTNV infection (Fig 14). These results may provide a basis for future investigation on the mechanism of NK cell immune response mediated by NKG2A-HLA-E axis, which may bring unique benefits for clinical treatment of HFRS. The inclusion of a limited number of cases considering the ongoing physical condition of HFRS patients placed certain limitations on the study. The next step will be to expand the sample size to ensure more accurate statistical results.

## Materials and methods

### Ethical statements

The formed consent was obtained from each HFRS patient or their guardian under a protocol approved by the Institutional Review Board of the Tangdu Hospital, the Xi'an Eighth Hospital, and the Fourth Military Medical University. The verbal formed consent to participate in this study was provided by child participants' legal guardian/next of kin. The research involving human materials was also approved by the Ethical Review Board of the First Affiliated Hospital of Fourth Military Medical University (Xi'an, China) with the license number KY20183312-1, and the related information was used anonymously.

### Patients

A total of 40 HFRS patients infected with HTNV were enrolled in this study at Department of Infectious Diseases at Tangdu Hospital of the Fourth Military Medical University (Xi'an, China) and the Xi'an Eighth Hospital (Xi'an, China). HTNV infection was confirmed by the detection of HTNV-specific immunoglobulin M (IgM) or IgG antibodies in serum specimens. Twenty-two healthy donors were enrolled as uninfected controls, showing anti-HTNV negative or no HTNV risk factors.

According to the diagnostic criteria from the Prevention and Treatment Strategy of HFRS promulgated by the Ministry of Health in the People's Republic of China, HFRS patients can be classified into four clinical types: mild, moderate, severe, and critical. Mild HFRS patients are diagnosed as mild renal failure without an obvious oliguric stage. Moderate HFRS patients show symptoms such as obvious symptoms of uremia, effusion (bulbar conjunctiva), hemorrhage (skin and mucous membrane) and renal failure with a typical oliguric stage. Severe HFRS patients are diagnosed as severe uremia, effusion (bulbar conjunctiva and either peritoneum or pleura), hemorrhage (skin and mucous membrane) and renal failure with oliguria (urine output 50-500 mL/day) for ≤ 5 days or anuria (urine output < 50 mL/day) for ≤ 2 days. Critical patients typically show more than one of the following symptoms: refractory shock, visceral hemorrhage, heart failure, pulmonary edema, brain edema, severe secondary infection and severe renal failure with oliguria (urine output 50-500 mL/day) for >5

days, anuria (urine output < 50 mL/day) for > 2 days, or a blood urea nitrogen (BUN) level of > 42.84 mmol/L. In this case, the number of patients with a severity degree of mild, moderate, severe, and critical was 4, 8, 11 and 7, respectively. To ensure the sample size in some statistical analyses, the HFRS patients with different severity could be divided into mild/moderate group and severe/critical group.

The clinical course of HFRS usually goes through five sequential stages: febrile, hypotensive, oliguric, diuretic and convalescent, which could be roughly divided into two phases including acute phase (febrile, hypotensive and oliguric stages) and convalescent phase (diuretic and convalescent stages). In general, samples were collected at 3-6 days for the febrile or hypotension stage, 7-12 days for the oliguric stage, after 13-18 days for the diuretic stage and after 18 days for the convalescent stage. In addition, patients with viral hepatitis, hematological diseases, autoimmune diseases, cardiovascular diseases or other kidney disease were excluded from this study. The general clinical information of the enrolled HFRS patients were provided in S1 Table.

### Sample collection

Peripheral blood samples of patients with HFRS were collected at acute phase and convalescence phase, respectively. Peripheral blood (10 mL) from each patient with HFRS and health donor was sterile extracted respectively with ethylenediaminetetraacetic acid (EDTA) anticoagulant. Following centrifugation at 2000 revolutions per minute (rpm) for 20 minutes (min), the plasma layer was collected and stored at -80°C. PBMCs were isolated by standard Ficoll-Hypaque density gradient centrifugation from anticoagulant peripheral blood and then resuspended with 10% fetal calf serum (FCS) RPMI 1640 for later experiments or frozen (90% FCS, 10% dimethyl sulfoxide) in liquid nitrogen for reserve. Meanwhile, the general information, clinical symptoms, clinical parameters and treatment plan of each patient were regularly recorded during the hospitalization of each patient.

### Single-cell RNA sequencing

The scRNA-Seq was performed by the Novel Bioinformatics Co., Ltd (Shanghai, China). Briefly, scRNA-Seq libraries were generated using the 10X Genomics Chromium Controller Instrument and Chromium Single Cell 5' library & gel bead kit (10X Genomics, Pleasanton, CA). The PBMCs of each HFRS patient or uninfected control were concentrated to 1,000 cells/µL. About 6,000 cells of each sample were loaded into each channel to generate single-cell gel bead-in-emulsions (GEMs). Following reverse transcription, GEMs were broken and the barcoded complementary DNA (cDNA) were purified and amplified. The amplified barcoded cDNA was then used to construct 5' gene expression libraries. Afterwards, the amplified barcoded cDNA was fragmented and added with poly(A)-tail. Then, adapters were ligated to the poly(A)-tailed cDNA fragments to create DNA libraries. The final libraries were quantified using the Qubit High Sensitivity DNA assay (Thermo Fisher Scientific, Waltham, MA, USA). The size distribution of the libraries was determined using a High Sensitivity DNA chip on the Bioanalyzer 2,200 (Agilent Technologies, USA). All libraries were sequenced using the illumina sequencer (Illumina, San Diego, CA) on a 150 bp paired-end run.

### Single-cell RNA statistical analysis

scRNA-seq data analysis was performed by Novel Bioinformatics Co., Ltd on the NovelBrain Cloud Analysis Platform. The adaptor sequence was filtered and low-quality reads were removed to obtain clean data. Afterwards, the feature-barcode matrices were obtained by aligning reads to the human genome (GRCh38 Ensemble: version 91) using cellranger v3.1.0. Cells that contained over 200 expressed genes and had a mitochondrial unique molecular identifier (UMI) rate below 20% passed the cell quality filtering. Meanwhile, the mitochondrial genes were removed from the expression table.

Moreover, the Seurat package (version: 2.3.4) was used for cell normalization. The count data was scaled regressing for total UMI counts and mitochondrial read percentage. Principal component analysis (PCA) was constructed based on the scaled data with the top 2,000 highly variable genes. In addition, the top 10 principals were used for constructing the

t-SNE graph. Using the graph-based cluster method, the unsupervised cell cluster was obtained based on the PCA top 10 principals. Moreover, the marker genes were calculated using the Find All Markers function with the Wilcoxon-rank sum test algorithm utilizing the following criteria: 1. Log2-FoldChange (FC) differential expression > 0.25; 2. *p*-value < 0.05; 3. min.pct > 0.1. Finally, the cell types from PBMCs were selected for t-SNE analysis, graph-based clustering and marker analysis to obtain details on NK cell clusters.

## Pathway analysis

Based on gene annotation databases, genes expression enriched in different signaling pathways were analyzed. The differentially expressed genes (DEGs) were analyzed by Kyoto Encyclopedia of Genes and Genomes (KEGG) database to obtain the biological functions and signaling pathways involved in disease occurrence and development. Statistical analysis of pathways was calculated by Fisher's test. $p < 0.05$ were considered statistically significant. The significant enriched pathway entries with statistical differences were selected for further analysis.

## Peptide synthesis

The predicted HLA-E*0103-restricted HTNV nine-mer peptides were synthesized with 90% purity at least and assessed by high-performance liquid chromatography and mass spectrometry (QYAOBIO, Shanghai, ChinaPeptides Co., Ltd.). The peptide from the HLA-Cw*03 leading sequences aa3-aa11 (VMAPRTLIL), which has been well-described as HLA-E-stabilizing peptides, was synthesized and used as a positive control. The peptide from human melanoma antigen-encoding gene-1 (MAGE-1) with sequence EADPTGHSY as well as HCMV phosphoprotein (PP) 65 with sequence NLVPMVATV, which could not bind to HLA-E molecules, were synthesized and used as negative controls, respectively. All the peptides were stored at 1 mM concentration at -70°C without repeated freeze-thawing.

## K562 cell lines transfected with HLA-E molecule

The chronic human myelogenous leukemia cell line K562 with no MHC molecules expression was obtained from ATCC (CCL-243). We stably transfected the K562 cell line with single allele HLA-E*0103 to generate mono-allelic cell lines with lentivirus transfection methods. Briefly, the lentivirus vector pLL3.7 (Genechem, Shanghai, China) with HLA-E*0103 allele was constructed. Lentivirus was produced by transfection in HEK293T cells using Lipofectamine 2000 transfection reagent (Life Technologies, Carlsbad, USA). K562 cell line was transfected with enhanced green fluorescent protein (EGFP) as well as the lentivirus in the presence of 6 µg/mL polybrene (Sigma-Aldrich, St. Louis, USA). The stable transfected cells were screened by 400 µg/mL hygromycin B (Life Technologies, Carlsbad, USA) in complete RPMI-1640 medium (Gibco, Mauricio Minotta, USA). Stably transfected cells (further designated as K562/HLA-E cells) showed strong expression (>95%) of HLA-E*0103 according to FCM analysis.

## K562/HLA-E cell binding assay

The stable transformed K562/HLA-E cells ($5\times10^5$) were incubated with each indicated peptide (100 µM) and human β2-microglobulin (β2m, Prospec-Tany, USA, 2.5µg/ml) in serum-free Opti-MEM (Thermo Fisher, USA) for 16-18h at 26°C with 5% $CO_2$. Meanwhile, the K562/HLA-E cells incubated without peptide was used as blank control.

For assessment of HLA-E/peptide binding stability with pulse-chase experiments, K562/HLA-E cells were pulsed with peptides as mentioned above, then washed twice, and resuspended in Opti-MEM without peptide or FCS. The expression of HLA-E on the surface of K562/HLA-E cells was then determined by staining with APC-labeled anti-human HLA-E mAb (clone 3D12, BioLegend, USA) and detected by ACEA NovoExpress system (Agilent Technologies, USA). The results were presented as the fluorescent index (FI), which was determined as follows: FI= (mean APC fluorescence with the

given peptide-mean APC fluorescence without peptide)/ (mean APC fluorescence without peptide). FI ≥ 1 represented high-affinity peptides, indicating that the peptide could be stably combined with HLA-E on the surface of K562/HLA-E cells to increase the mean fluorescence of the HLA-E molecule by at least onefold.

Furthermore, the peptide binding affinity was assessed by detecting the expression level of HLA-E on K562/HLA-E cells at 0 min following incubation with each indicated peptide incubation under different peptide concentrations (1 μM, 10 μM, 30 μM, 50 μM, 100 μM, 200 μM and 300 μM). Meanwhile, the expression levels of HLA-E on K562/HLA-E cells were detected at indicated time points (0 min, 30 min, 60 min, 120 min, 240 min, 360 min and 480 min) after incubation with each of the peptide at 100 μM to evaluate the peptide binding stability with HLA-E. The leading sequence peptide from HLA-Cw*03 (VMAPRTLIL) was used as positive control peptide. Peptide from melanoma antigen-encoding gene (MAGE-1, EADPTGHSY) was used as negative control peptide.

### Ligand-receptor binding assay

Peptide-pulsed K562/HLA-E cells were firstly incubated with 4 μg/mL recombinant biotinylated CD94/NKG2A or CD94/NKG2C protein (Acro Biosystems, USA) for 20 min on ice. After washing, the cells were detection with Streptavidin-PE/Cyanine7 (BioLegend, USA), which could bind to biotinylated proteins with high affinity. Meanwhile, peptide-pulsed K562/HLA-E cells were stained with APC-labeled anti-human HLA-E mAb (clone 3D12, BioLegend, USA). Then the CD94/NKG2A+HLA-E+ and CD94/NKG2C+HLA-E+ K562/HLA-E cells were gated as targeted cells.

### Tetramer synthesis and HTNV-derived peptide/HLA-E*0103 tetramer staining

Two HTNV-derived peptides (NP5 and GP8) were selected for the construction of HTNV-derived peptide/HLA-E*0103 tetramer labeled with PE by Epigen Biotech (Nantong, China). Western blot was used to detect the efficiency of biotinylation for each HTNV-derived peptide/HLA-E*0103 tetramer. The binding rates of biotinylated HLA-E*0103 monomer to streptavidin for both the two peptides ≥ 80%. Ranking of affinity for the two peptides were HTNV-GP8 (MMKSCLIAL)> HTNV-NP5 (QLGQRIIVL).

For HTNV-derived peptide/HLA-E*0103 tetramer staining, PBMCs (1×10⁶) of HFRS patients and uninfected controls were stained with each PE-labeled HLA-E*0103 tetramer for 10 min at room temperature, and subsequently stained with Brilliant Violet 650-labeled anti-human CD3 mAb (Clone HIT3a, BioLegend, USA), Brilliant Violet 510-labeled anti-human CD56 mAb (Clone HCD56, BioLegend, USA), APC/Fire-labeled anti-human CD16 mAb (Clone 3G8, BioLegend, USA), PE-Cy7-labeled anti-human NKG2A mAb (Clone S19004C, BioLegend, USA), Brilliant Violet 786-labeled anti-human NKG2C mAb (Clone 134591, BD Biosciences, USA) and Brilliant Violet 605-labeled anti-human CD57 mAb (Clone QA17A04, BioLegend, USA) for 20 min at 4°C. Then the CD3-CD56dimCD16+NKG2A+NKG2C-CD57-tetramer+ NK cells were gated according to isotype control. Compensation controls were checked regularly to avoid false-positive results and individually determined for each experimental setup.

### Isolation of NK cells with magnetic bead kits

NK cells were isolated from the PBMCs of each HFRS patient and uninfected control respectively, using the human NK cell negative selection kit (Miltenyi Biotec, Bergisch Gladbach, Germany) according to the manufacturer's instructions. PBMCs were firstly incubated with a biotin-antibody cocktail at 4°C for 5 min. Without washing, the cells followed by adding the microBeads cocktail and incubating at 4°C for 10 min. The cell suspension was then applied onto the MACS column to collect flow-through containing unlabeled cells, representing the enriched NK cells. The sorted NK cells (5×10⁵) were then co-cultured with K562/HLA-E cells (5×10⁵) loaded with the indicated peptide at 37°C for 4 h. Subsequently, peptide loaded-K562/HLA-E cells co-cultured NK cells were collected for surface staining, intracellular cytokine staining and degranulation assay.

## Surface staining

PBMCs ($1\times10^6$) or peptide loaded-K562/HLA-E cells co-cultured NK cells ($5\times10^5$) from each HFRS patient and uninfected control were stained with different fluorescence labelled-antibodies specific to CD3, CD56, CD16, NKG2A, NKG2C, CD57 as mentioned above, as well as stained with activation marks Brilliant Violet 421-labeled anti-human CD38 mAb (Clone HIT2, BioLegend, USA), APC-labeled anti-human HLA-DR mAb (Clone L243, BioLegend, USA) and FITC-labeled anti-human CD69 mAb (Clone FN50, BioLegend, USA) for 30 min at 4°C. A minimum of 500,000 total cells were acquired and gated on the CD3-CD56dimCD16+NKG2A+NKG2C-CD57- NK cells. The activation marks CD38, HLA-DR and CD69 were further analyzed. The ACEA NovoExpress system (Agilent Technologies, USA) was used for data acquisition and analysis.

## Intracellular cytokine staining and degranulation assay

PBMCs ($1\times10^6$) or peptide loaded-K562/HLA-E cells co-cultured NK cells ($5\times10^5$) from each patient and uninfected control were stimulated with phorbol myristate acetate (PMA, 20 μg/mL, eBiosciences, USA), ionomycin (1.5 μM, eBiosciences, USA) and monensin (Golgistop, 1.5 μM, eBiosciences, USA) at 37°C for 4 h. For the detection of degranulation responses to antigen stimulation, the APC-labeled anti-human CD107a mAb (Clone H4A3, BioLegend, USA) was added to the cells during stimulation. Cells were collected and stained with different fluorescence labelled-antibodies against the surface markers, including CD3, CD56, CD16, CD57, NKG2A and NKG2C. Then Foxp3/Transcription Factor Fixation/Permeabilization kit (eBiosciences, USA) was employed for cell treatment. Cells were subsequent stained with antibodies specific to intracellular markers, including FITC-labeled anti-human IFN-γ mAb (Clone 4S.B3, BioLegend, USA), Brilliant Violet 421-labeled anti-human TNF-α (Clone MAb11, BioLegend, USA), APC-labeled anti-human Ki-67 (Clone Ki-67, BioLegend, USA), FITC-labeled anti-human granzyme B (Clone QA16A02, BioLegend, USA) and Brilliant Violet 421-labeled anti-human perforin (Clone dG9, BioLegend, USA) for 30 min at 4°C, followed by acquisition using the ACEA NovoExpress system (Agilent Technologies, USA). NK cells were defined as CD3-CD56dimCD16+NKG2A+NKG2C-CD57- events, which were further analyzed for expression of Ki-67, IFN-γ, TNF-α, CD107a, perforin and granzyme B.

## Isolation of NK cells with FACS

PBMCs ($3\times10^7$) from uninfected controls were stained with PE-Cy7-labeled anti-human CD3 mAb (Clone 17A2, BioLegend, USA), PE-labeled anti-human CD56 mAb (Clone 5.1H11, BioLegend, USA), and APC-labeled anti-human NKG2A mAb (Clone S19004C, BioLegend, USA). The cells were incubated for 30 minutes at 4°C in the dark. After incubation, the cells were washed and resuspended in RPMI-1640 medium (Gibco, USA). The CD3- CD56dimNKG2A+ NK cells and CD3- CD56dimNKG2A- NK cells were sorted using a BD FACS Aria IIU system (BD Biosciences, USA). The sorted cells were then rested in complete RPMI-1640 medium supplemented with 200 U/ml IL-2 (MedChemExpress, USA) overnight at 37°C and 5% $CO_2$ before the start of the killing assay.

## NK cell cytotoxicity assay

The CD3- CD56dimNKG2A+NK cells and CD3- CD56dimNKG2A-NK cells, sorted by FACS from PBMCs of uninfected controls, were used as effector cells. EGFP-labeled K562/HLA-E cells pulsed with HTNV-derived peptides (NP5, GP7, GP8, and GP20) were used as target cells. The effector cells were co-cultured with $1\times10^4$ peptide-loaded K562/HLA-E target cells in a 96-well U-bottom plate at an effector-to-target (E:T) ratio of 3:1. Control groups included NK cells only and EGFP-labeled K562/HLA-E cells only. After a 5-hour incubation at 37°C, cells were stained for viability using Fixable Viability Dye eFluor 450 (BioLegend, USA) at the manufacturer's recommended concentration for 30 minutes at 4°C. The cells were then analyzed by flow cytometry using an ACEA NovoExpress system (Agilent Technologies, USA). EGFP+ Fixable Viability Dye eFluor 450+ cells were identified as killed target cells.

## Statistical analysis

The statistical analysis was performed using Prism software, version 8.0 (Graphpad, La Jolla, CA, USA). The frequencies of activation, proliferation, cytokine and cytotoxic mediator production of NK cells were presented as the medians and range values. The Mann-Whitney U test was used for parameter comparison between two subject groups. All the statistical tests were two sided and $p$-values < 0.05 were considered significant.

## Supporting information

**S1 Fig. Comparison expression of selected genes (CD56/*NCAM1*, CD16/*FCER1G*, CD57/*B3GAT1*, NKG2A/*KLRC1* and NKG2C/*KLRC2*) of peripheral blood NK cells.** The violin graph showing the comparison of the above genes expression in total NK cells (A) between uninfected controls and HFRS patients; and (B) in HFRS patients with different disease severities and uninfected controls.
(TIF)

**S2 Fig. Gating strategy for CD56$^{dim}$CD16$^+$NKG2A$^+$NKG2C$^-$CD57$^-$ NK cells.** Based on the expression of CD3, CD56, CD16, NKG2A, NKG2C, CD57, NK cells were gated as CD3$^-$CD56$^{dim}$CD16$^+$NKG2A$^+$NKG2C$^-$CD57$^-$ cells (Cells with the above phenotype were referred to as CD56$^{dim}$NKG2A$^+$ NK cells, which were the target subgroup of the research).
(TIF)

**S3 Fig. The comparison of cytotoxic mediators production between CD56$^{dim}$NKG2A$^+$ and CD56$^{dim}$NKG2A$^-$ NK cells in uninfected controls.** Representative flow cytometric plots and the comparison of the frequencies of perforin (A and B), granzyme B (C and D) and the expression of CD107a (E and F) on CD56$^{dim}$CD16$^+$NKG2A$^{+/-}$NKG2C$^-$CD57$^-$ NK cells.
(TIF)

**S1 Table. Characteristics of enrolled HFRS patients.**
(XLSX)

**S1 Data. This Excel file contains the source data of all main and supplementary figures.** Each sheet in the file contains the data for one figure and is labelled accordingly.
(XLSX)

## Acknowledgments

We thank the generous volunteers who participated in this study.

## Author contributions

**Conceptualization:** Xuyang Zheng, Ying Ma.

**Data curation:** Kang Tang, Yusi Zhang.

**Formal analysis:** Manling Xue, Xiaoyue Xu, Jiajia Zuo.

**Funding acquisition:** Kang Tang, Yusi Zhang, Chunmei Zhang, Yun Zhang, Ying Ma.

**Investigation:** Chunmei Zhang.

**Methodology:** Manling Xue.

**Project administration:** Ying Ma.

**Resources:** Ying Ma.

**Software:** Fenglan Wang.

**Supervision:** Ran Zhuang, Yun Zhang.

**Validation:** Xiyue Zhang.

**Visualization:** Xuyang Zheng.

**Writing – original draft:** Manling Xue.

**Writing – review & editing:** Boquan Jin, Ying Ma.

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
