## [Decision Letter · Decision Letter 0]

PPATHOGENS-D-24-02216

Hantaan virus-derived peptides that stabilize HLA-E could abrogate inhibition of CD56dimNKG2A+ NK cells

PLOS Pathogens

Dear Dr. Ma,

Thank you for submitting your manuscript to PLOS Pathogens. After careful consideration, we feel that it has merit but does not fully meet PLOS Pathogens's publication criteria as it currently stands. Therefore, we invite you to submit a revised version of the manuscript that addresses the points raised during the review process.

Please submit your revised manuscript within 60 days Apr 21 2025 11:59PM. If you will need more time than this to complete your revisions, please reply to this message or contact the journal office at plospathogens@plos.org. Please include the following items when submitting your revised manuscript:

We look forward to receiving your revised manuscript.

Kind regards,

William J. Liu

Guest Editor

PLOS Pathogens

Matthias Schnell

Section Editor

PLOS Pathogens

 Sumita Bhaduri-McIntosh

Editor-in-Chief

PLOS Pathogens

orcid.org/0000-0003-2946-9497

 Michael Malim

Editor-in-Chief

PLOS Pathogens

orcid.org/0000-0002-7699-2064

**Additional Editor Comments:**

As the comments from the reviewers, the study offers valuable insights into how HTNV-derived peptides interact with NK cells and impact the function, which could inform future therapeutic developments. However, some concerns are also raised by the reviewers. The following comments and suggestions from the reviewers may benefit the revision of the work for the authors.

1. The authors need to make clearer analysis and explanation for the following results, such as the unusual detection of MHC class II molecular HLA-DR expression, high CD69 expression in NK cells of HFRS patients at convalescence stage. Also, the author should explain the high expression of perforin, granzyme B, and CD107a in the uninfected control group, and the inconsistent expression profiles of TNF-a, Perforin, and CD107a in Fig.11 compared to Fig.6 and Fig.7. Further, the statistical analysis of the increase of NK cell cytotoxic capacity in the presence of HTNV-derived peptide in Fig.13 need to be enhanced to support the conclusion.

2. The sample size is relatively small. This limits the statistical power of the study. The authors should clearly state the limitations in the discussion, and also tone down the conclusion about the correlations between NK cell characteristics and disease severity.

3. As the suggestion by two reviewers, to explain the specific effect of peptide NP5, a peptide known to induce an inhibitory effect on NK cells may be needed.

**Journal Requirements:**

At this stage, the following Authors/Authors require contributions: Xue Manling, Tang Kang, Zhang Yusi, Xu Xiaoyue, Zhang Chunmei, Zuo Jiajia, Wang Fenglan, Zhang Xiyue, Zheng Xuyang, Zhuang Ran, Zhang Yun, Jin Boquan, and Ying Ma. Please ensure that the full contributions of each author are acknowledged in the "Add/Edit/Remove Authors" section of our submission form.

2) We note that your Manuscript "Hantaan virus-derived peptides.pdf" and Manuscript " Manuscript.pdf" files are duplicated on your submission. Please remove any unnecessary or old files from your revision, and make sure that only those relevant to the current version of the manuscript are included.

3) We ask that a manuscript source file is provided at Revision. Please upload your manuscript file as a .doc, .docx, .rtf or .tex. If you are providing a .tex file, please upload it under the item type u2018LaTeX Source Fileu2019 and leave your .pdf version as the item type u2018Manuscriptu2019.

4) Please ensure that the funders and grant numbers match between the Financial Disclosure field and the Funding Information tab in your submission form. Note that the funders must be provided in the same order in both places as well. State what role the funders took in the study. If the funders had no role in your study, please state: "The funders had no role in study design, data collection and analysis, decision to publish, or preparation of the manuscript.".

**Reviewers' Comments:**

Reviewer's Responses to Questions

**Part I - Summary**

Reviewer #1: Xue et al investigated the NK cell responses in HFRS patients cause by Hantaan virus (HTNV) infection. They described that several HTNV-derived peptides can stabilize HLA-E expression on the target cells, but the HLA-E/HTNV-derived peptide complex are not recognized by CD94/NKG2A receptor on NK cells, which may enhance the NK cell activation during the virus infections. However, the increase of NK cell cytotoxic capacity in the presence of HTNV-derived peptide are neglectable. Many of the data in this manuscript have inconsistencies with each other.

Reviewer #2: This manuscript, titled "Hantaan Virus-Derived Peptides That Stabilize HLA-E Could Abrogate Inhibition of CD56dimNKG2A+ NK Cells," investigates the role of Hantaan virus (HTNV)-derived peptides in modulating the activity of CD56dimNKG2A+ natural killer (NK) cells in patients with hemorrhagic fever with renal syndrome (HFRS). The study employs a combination of single-cell RNA sequencing, flow cytometry, and functional assays to characterize NK cell subsets and thoroughly examines their responses to HTNV peptides. The findings indicate that specific HTNV peptides presented by HLA-E can counteract the inhibitory effect of NKG2A, thereby enhancing antiviral immunity. Additionally, the study offers valuable mechanistic insights into how HTNV-derived peptides may evade NK cell-mediated inhibition, which could inform future therapeutic developments. While the study delivers intriguing results and demonstrates strong methodology, further clarification and improvements are required.

In summary, this study offers valuable insights into the interactions among HTNV, HLA-E, and NK cells. However, it has notable limitations, including a very small sample size, the necessity for additional experiments to confirm the observed correlations, and a lack of definitive mechanistic details. Addressing these concerns will significantly enhance the manuscript's robustness and increase its overall impact.

Reviewer #3: The authors investigated the role of NK cells in Hantaan virus induced hemorrhagic fever with renal syndrome (HFRS). NK subtypes in HFRS patients were quantified and none of 4 virus peptides that could be presented by HLA-E was recognized by NKG2A, suggesting that a lack of recognition might leading to enhanced NK cell activity by not triggering the NK cell inhibitory activity mediated by NKG2A. The results are somewhat convincing but strictly correlative.

**Part II – Major Issues: Key Experiments Required for Acceptance**

Reviewer #1: 1. It seems that the 0 cluster of NK cells are the most abundant subsets of NK cells in HFRS patients, which express lower level of NKG2A, CD16 and CD56. What is the function and activation status of the cluster 0 in HFRS patients?

2. In Figure 4, it is unusual to detect MHC class II molecular HLA-DR expression in NK cells. It is also unusual to detect high CD69 expression in NK cells of HFRS patients at convalescence stage. The author should explain this.

3. In Figure 7, The trends of perforin, granzyme B, and CD107a in the same cells were not consistent. Granzyme B and CD107a should have similar trends under normal circumstances.

4. Surprisingly, in the uninfected control group, the expression of perforin, granzyme B, and CD107a was so high (Fig. 7), the authors should use CD56dimCD16+NKG2A- NK cells as a control to compared the expression profiles of perforin, granzyme B, and CD107a between NKG2A+ and NKG2A- NK cells.

5. The expression profiles of TNF-a, Perforin, and CD107a in Fig.11 were not consistent with that in Fig.6 and Fig.7.

6. The increase of NK cell cytotoxic capacity in the presence of HTNV-derived peptide were marginal in Fig.13, cannot support the conclusion.

Reviewer #2: • The sample size (six HFRS patients and two controls) is relatively small. This limits the statistical power of the study and raises concerns about the generalizability of the findings. The authors should consider increasing the sample size. Otherwise, please clearly state the limitations due to the small sample size in the discussion.

• The study demonstrates correlations between NK cell characteristics and disease severity. However, more work is needed to definitively establish causal relationships. The authors should design experiments to further validate the findings.

• While the study shows that HTNV peptides abrogate NKG2A inhibition, the precise molecular mechanisms underlying this abrogation require further investigation. The authors should explore further the interaction dynamics between HLA-E/peptide complexes and CD94/NKG2A.

• The study focuses primarily on one HTNV peptide (NP5). Further investigation with other identified peptides is necessary to establish whether this effect is specific to NP5 or a common property of multiple HTNV peptides.

• The authors should expand the control groups to include healthy individuals without evidence of HTNV exposure or other relevant viral infections to better account for confounding factors.

Reviewer #3: Figure 3 - NKG2A is higher in cells from HFRS patients. What is mediating this and could it simply be the higher amounts of NKG2A and not its engagement by peptide bearing HLA-E is driving the enhanced disease?

Figure 6 legend and lines 248-250 - Since the cytokines analyzed are IFNg and TNFa , the word "cytokines" should be replaced with "IFNg and TNFa" unless the authors are willing to look for more cytokines.

Figure 11 - Can the authors use a peptide known to induce an inhibitory effect on NK cells in this experiment to show that inhibition of NK cell activation is possible in this assay?

**Part III – Minor Issues: Editorial and Data Presentation Modifications**

Reviewer #1: The manuscript was poorly written, with many writing errors, such as “Patents” in line 604; “CD3D, FCER1G, FCERIG, NCR1” in line 123. The figure legends need more details about how they performed the experiments.

Reviewer #2: • Some figures are complex and could benefit from clearer labeling, improved visual presentation, and more concise legends. The authors should consider consolidating or simplifying some of the figures.

Reviewer #3: Lines 26-29 - There is a lot speculation in this sentence.

With all six infected patients being males but one of the two uninfected patients being female, it is important for the authors to make some statements about sex-differences in HFRS and how this skewing of their study populations might impact the significance of their results.

PLOS authors have the option to publish the peer review history of their article (what does this mean? ). If published, this will include your full peer review and any attached files.

**Do you want your identity to be public for this peer review?** For information about this choice, including consent withdrawal, please see our Privacy Policy .

Reviewer #1: No

Reviewer #2: No

Reviewer #3: No

**Figure resubmission:**
---

## [Decision Letter · Decision Letter 1]

Dear Dr Ma,

We are pleased to inform you that your manuscript 'Hantaan virus-derived peptides that stabilize HLA-E could abrogate inhibition of CD56dimNKG2A+ NK cells' has been provisionally accepted for publication in PLOS Pathogens.

Best regards,

William J. Liu

Guest Editor

PLOS Pathogens

Matthias Schnell

Section Editor

PLOS Pathogens

Sumita Bhaduri-McIntosh

Editor-in-Chief

PLOS Pathogens

orcid.org/0000-0003-2946-9497

Michael Malim

Editor-in-Chief

PLOS Pathogens

orcid.org/0000-0002-7699-2064

The authors addressed all the concerns raised by the reviewers, especially the issues that I emphasized in the last decision. And all the three reviewers gave the recommendation of acceptance for the current version.

Reviewer Comments (if any, and for reference):

Reviewer's Responses to Questions

**Part I - Summary**

Reviewer #1: The authors have satisfactorily addressed my concerns. I have no further comments.

Reviewer #2: The authors have clearly addressed previous reviewer comments and editor suggestions, particularly regarding statistical robustness, sample size limitations, and mechanistic explanations. In addition, the revised manuscript includes more detailed explanations of the methods used (flow cytometry, cell sorting, etc.).

Reviewer #3: The authors were responsive to the main critiques and have improved the manuscript data and discussion.

**Part II – Major Issues: Key Experiments Required for Acceptance**

Reviewer #1: (No Response)

Reviewer #2: No

Reviewer #3: (No Response)

**Part III – Minor Issues: Editorial and Data Presentation Modifications**

Reviewer #1: (No Response)

Reviewer #2: No

Reviewer #3: (No Response)

PLOS authors have the option to publish the peer review history of their article (what does this mean? ). If published, this will include your full peer review and any attached files.

**Do you want your identity to be public for this peer review?** For information about this choice, including consent withdrawal, please see our Privacy Policy .

Reviewer #1: No

Reviewer #2: **Yes: ** Xi Wang

Reviewer #3: No

---

## [Editor Report · Acceptance letter]

Dear Dr Ma,

We are delighted to inform you that your manuscript, "Hantaan virus-derived peptides that stabilize HLA-E could abrogate inhibition of CD56dimNKG2A+ NK cells," has been formally accepted for publication in PLOS Pathogens.

Best regards,

Sumita Bhaduri-McIntosh

Editor-in-Chief

PLOS Pathogens

orcid.org/0000-0003-2946-9497

Michael Malim

Editor-in-Chief

PLOS Pathogens

orcid.org/0000-0002-7699-2064